# Coopted temporal patterning governs cellular hierarchy, heterogeneity and metabolism in *Drosophila* neuroblast tumors

Sara Genovese[1], Raphaël Clément[1†], Cassandra Gaultier[1†], Florence Besse[2], Karine Narbonne-Reveau[1], Fabrice Daian[1], Sophie Foppolo[1], Nuno Miguel Luis[1], Cédric Maurange[1*]

[1]Aix Marseille Univ, CNRS, IBDM, Equipe Labellisée LIGUE Contre le Cancer, Marseille, France; [2]Université Côte d'Azur, CNRS, Inserm, Institut de Biologie Valrose, Nice, France

*For correspondence:
cedric.maurange@univ-amu.fr

[†]These authors contributed equally to this work

Competing interests: The authors declare that no competing interests exist.

## Abstract

It is still unclear what drives progression of childhood tumors. During *Drosophila* larval development, asymmetrically-dividing neural stem cells, called neuroblasts, progress through an intrinsic temporal patterning program that ensures cessation of divisions before adulthood. We previously showed that temporal patterning also delineates an early developmental window during which neuroblasts are susceptible to tumor initiation (Narbonne-Reveau et al., 2016). Using single-cell transcriptomics, clonal analysis and numerical modeling, we now identify a network of twenty larval temporal patterning genes that are redeployed within neuroblast tumors to trigger a robust hierarchical division scheme that perpetuates growth while inducing predictable cell heterogeneity. Along the hierarchy, temporal patterning genes define a differentiation trajectory that regulates glucose metabolism genes to determine the proliferative properties of tumor cells. Thus, partial redeployment of the temporal patterning program encoded in the cell of origin may govern the hierarchy, heterogeneity and growth properties of neural tumors with a developmental origin.
DOI: https://doi.org/10.7554/eLife.50375.001

## Introduction

Central nervous system (CNS) tumors are rare and constitute less than 2% of all cancers in adults. In contrast, they represent more than 25% of cancer cases in children (including medulloblastoma, retinoblastoma, rhabdoid tumors (AT/RT), gliomas etc), suggesting that the developing CNS is particularly sensitive to malignant transformation (*Arora et al., 2009*; *Curado et al., 2007*). Moreover, unlike most adult tumors, pediatric tumors are often genetically stable and their initiation and progression do not necessarily require the accumulation of mutations in multiple genes. For example, the biallelic inactivation of a single gene is sometimes sufficient to trigger malignant growth as illustrated by mutations in the RB1 and SMARCB1 genes in retinoblastoma and rhabdoid tumors respectively (*Biswas et al., 2016*; *Gröbner et al., 2018*; *Marshall et al., 2014*; *Puisieux et al., 2018*; *Scotting et al., 2005*; *Vogelstein et al., 2013*). Recent studies suggest that CNS pediatric tumors such as medulloblastomas recapitulate the fetal transcription program that was active in the cell of origin (*Vladoiu et al., 2019*). However, it remains unclear how the invalidation of single genes during fetal stages can disrupt on-going developmental programs to trigger malignant growth, and whether these fetal/developmental programs influence the heterogeneity, composition, and proliferative properties of cells composing CNS tumors.

Faced with the complexity of brain development and neural tumors in mammals, simple animal models can represent a powerful alternative to investigate basic and evolutionary conserved principles. The development of the CNS is undoubtedly best understood in *Drosophila* (*Homem and Knoblich, 2012*). The *Drosophila* CNS arises from a small pool of asymmetrically-dividing neural stem cells (NSCs), called neuroblasts (NBs). NBs possess a limited self-renewing potential. They divide all along development (embryonic and larval stages) to self-renew while generating daughter cells named Ganglion Mother Cells (GMCs) (*Maurange and Gould, 2005*). GMCs then usually divide once to produce two post-mitotic neurons or glia. NBs are the fastest cycling cells during development, able to divide every hour during larval stages when most of the neurons are produced (*Truman and Bate, 1988*). However, all NBs terminate during metamorphosis and are absent in adults. Two antagonistic RNA-binding proteins, IGF-II mRNA-binding protein (Imp) and Syncrip (Syp) are essential to first promote and then conclude this formidable period of activity. During early larval development (L1/L2), NBs express Imp that promotes NB self-renewal. Around late L2/early L3, NBs silence Imp to express Syp that remains expressed until NB decommissioning during metamorphosis (*Yang et al., 2017*). This Imp-to-Syp transition is essential to render NBs competent to respond to subsequent pupal pulses of the steroid hormone ecdysone and initiate a last differentiative division (*Homem et al., 2014*; *Yang et al., 2017*). Failure to trigger the transition results in NBs permanently dividing in adults (*Maurange et al., 2008*; *Narbonne-Reveau et al., 2016*; *Yang et al., 2017*). The Imp-to-Syp transition appears to be mainly regulated by a NB intrinsic timing mechanism driven by the sequential expression of transcription factors (*Narbonne-Reveau et al., 2016*; *Ren et al., 2017*; *Syed et al., 2017*). This series of factors, also known as temporal transcription factors, has been first identified for its ability to specify different neuronal fates produced by NBs as they divide (*Bayraktar and Doe, 2013*; *Isshiki et al., 2001*; *Li et al., 2013*). In addition, temporal transcription factors also schedule the Imp-to-Syp transition to ensure that NBs will not continue cycling in adults. Recent transcriptomic analyses indicate that other genes are dynamically transcribed in NBs throughout larval stages, although their function and epistatic relationship with temporal transcription factors and the Imp/Syp module are unclear (*Liu et al., 2015*; *Ren et al., 2017*; *Syed et al., 2017*) All together, these studies highlight a complex, but still relatively unexplored, temporal patterning system in larval NBs.

Perturbation of the asymmetric division process during early development can lead to NB exponential amplification. In such conditions, the NB-intrinsic temporal program limiting self-renewal appears to become inoperant, and uncontrolled NB amplification is observed. Serial transplantations of asymmetric division-defective NBs have revealed an ability to proliferate for months, if not years, demonstrating tumorigenic characteristics (*Caussinus and Gonzalez, 2005*). Perturbation of asymmetric divisions can be induced by the inactivation of the transcription factor Prospero (Pros) in type I NB lineages (most lineages in the ventral nerve cord (VNC) and central brain (CB)). During development, Pros is strongly expressed in GMCs where it accumulates to induce cell cycle-exit and neuronal or glial differentiation (*Choksi et al., 2006*; *Hirata et al., 1995*; *Matsuzaki et al., 1992*). GMCs that lack *pros* fail to differentiate and revert to a NB-like state. This triggers rapid NB amplification at the expense of neuron production (*Bello et al., 2006*; *Caussinus and Gonzalez, 2005*). We have previously shown that inactivation of *pros* in NBs, and their subsequent GMCs, before mid-L3 (L3 being the last larval stage) leads to aggressive NB tumors that persist growing in adults. In contrast, inactivation of *pros* after mid-L3 leads to transient NB amplification and most supernumerary NB properly differentiate during metamorphosis, leading to an absence of growing tumors in adults (*Narbonne-Reveau et al., 2016*). Interestingly, propagation of NB tumor growth beyond normal developmental stages is caused by the aberrant maintenance of Imp and the transcription factor Chinmo from early-born GMCs, the latter representing the cells of origin of such aggressive tumors (*Narbonne-Reveau et al., 2016*). Chinmo and Imp positively cross-regulate and inactivation of either in NB tumors stops tumorigenic growth. Because *pros*[-/-] NB tumors can only be induced during an early window of development, and are caused by the biallelic inactivation of a single gene, they represent an exciting and simple model to investigate the basic mechanisms driving the growth of tumors with an early developmental origin, such as in the case of pediatric CNS cancers.

NB tumors can also be induced from type II NBs (a small subset of NBs in the central brain) or from neurons upon inactivation of the NHL-domain family protein Brat or Nerfin-1 respectively (*Bello et al., 2006*; *Betschinger et al., 2006*; *Lee et al., 2006*) (*Froldi et al., 2015*). In both cases, tumor growth appears to rely on the aberrant expression of the Chinmo/Imp module arguing for a general tumor-driving mechanism in the developing *Drosophila* CNS (*Narbonne-Reveau et al., 2016*). Interestingly, in the different types of NB tumors, Chinmo and Imp are only expressed in a subpopulation of cells, demonstrating heterogeneity in the population of tumor NBs (tNBs). However, the full repertoire of cells composing the tumor, the rules governing the cellular heterogeneity and the mechanisms determining the proliferative potential of each cell type remain to be investigated.

Here, we use single-cell RNA-seq, clonal analysis and numerical modeling to investigate these questions. We identify a subset of genes involved in the temporal patterning of larval NBs that are redeployed in tumors to generate a differentiation trajectory responsible for creating tumor cell heterogeneity. This cellular heterogeneity results in NBs with different types of metabolism and different proliferative properties. We also decipher a robust hierarchical scheme that drives reproducible heterogeneity through the dysregulated but fine-tuned transition between the two RNA-binding proteins Imp and Syp. This work thus identifies a core larval NB temporal patterning program, the disruption of which not only causes unlimited growth but has an overarching role in governing the cellular hierarchy, heterogeneity and metabolism of NB tumors.

## Results

### Single-cell RNA-seq reveals that a subset of larval NB temporal patterning genes are major contributors of cellular heterogeneity within NB tumors

To investigate the cellular heterogeneity existing in a given type of NB tumor, we performed single-cell RNA-seq experiments. To minimize inter-tumor heterogeneity that could result from inducing tumors with different NBs of origin, tumors were induced from the early L2 stage, when NBs exit quiescence, by knocking down *pros* in the six homologous $pox^n$ NBs of the VNC (one per hemisegment) using the $pox^n$-*Gal4, UAS-pros$^{RNAi}$, UAS-GFP, UAS-dicer2* system - thereafter referred to as $pox^n > pros^{RNAi}$. We had previously demonstrated that early NB amplification triggered during early larval stages by the $pox^n > pros^{RNAi}$ system led to tumors that persist and expand in adults (*Narbonne-Reveau et al., 2016*). Single-cell RNA-seq was performed on dissociated and FACS-sorted GFP$^+$ tNBs obtained from 56 $pox^n > pros^{RNAi}$ tumors dissected and pooled from adults aged between 4 to 6 days (tumors were therefore induced 11 to 13 days earlier).

Using the Chromium (10x Genomics) approach, we sequenced 5740 cells with a median number of 1806 genes/cell (*Figure 1—figure supplement 1A*). Filtering, normalization, variable gene identification, regression of cell-cycle genes, linear dimensional reduction, and principal component analysis (PCA) were performed with the R package Seurat (*Butler et al., 2018*) (*Figure 1—figure supplement 1A,B*). PC3, PC4 and PC7 in particular caught our attention, as they all identify the early and late larval NB temporal markers, Imp and Eip93F (E93) (*Syed et al., 2017*), being expressed in different subsets of tNBs (*Figure 1A* and *Figure 1—figure supplement 1C*). This indicates that early and late larval NB markers are major components defining cellular heterogeneity in NB tumors.

We next proceeded to graph-based clustering. This led to a UMAP representation composed of five main clusters and two smaller ones (*Figure 1B*). Although tNBs $pox^n > pros^{RNAi}$ tumors may occasionally differentiate in neurons (*Narbonne-Reveau et al., 2016*), $pox^n$-*GAL4* is only active in the tNB population. Therefore, we expected the population of GFP$^+$ cells after FACS sorting to be exclusively composed of tNBs. Consistently, NB identity genes like *miranda* (*mira*) or *deadpan* (*dpn*), appeared homogeneously distributed on the UMAP representation while genes defining cholinergic (*VAChT*), GABAergic (*Gad1*) and glutamatergic (*VGlut*) neurons were absent (*Figure 1C* and *Figure 1—figure supplement 2A*). Similarly, pan-glial genes (*repo*) were not detected (*Figure 1—figure supplement 2A*). The two smaller clusters appear to be composed of tNBs expressing stress or growth arrest factors such as *Gadd45*, *Irbp18*, *Xrp1* and *GstE6* possibly composing a sub-population of tNBs under cellular stress (*Figure 1—figure supplement 2B*) (*Akdemir et al., 2007*; *Francis et al., 2016*; *Lee et al., 2018*). Differential expression analysis identifies 'cluster 2' (1268 out

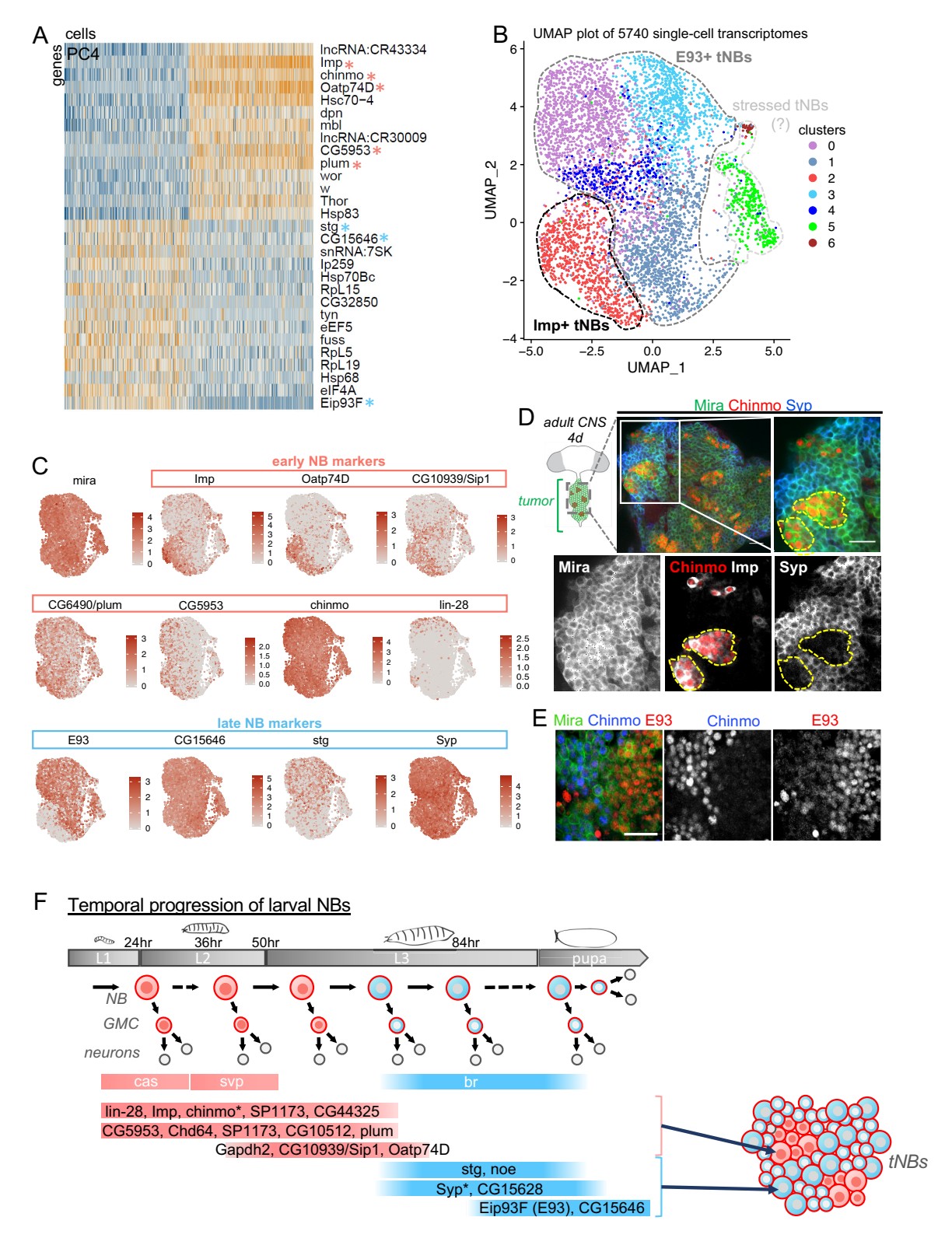

**Figure 1.** Single-cell RNA-seq identifies a subset of temporal patterning genes that are redeployed in NB tumors defining heterogeneity in NB states. (**A**) PCA was performed on single-cell tNB transcriptomes to reduce the dimensions of the data for further analysis. Genes (rows) and cells (columns) are ordered by their PCA scores, and the 500 most extreme cells and 30 most extreme genes on both sides of the distribution are shown in the heatmap. PC4 reveals that tNBs can be discriminated by the expression of early (light red asterisks) vs late (light blue asterisks) larval NB temporal patterning

*Figure 1 continued on next page*

*Figure 1 continued*

genes. Other larval temporal genes are found in PC3 and PC7 (*Figure 1—figure supplement 1C*). (B) The UMAP representation of all single cells included in our analysis shows the separation of different clusters. We used a k-nearest neighbor algorithm to call seven clusters, which are shown in different colors on the UMAP plot. (C) Expression of Mira, early and late temporal markers on the UMAP map shown in B. (D) Cartoon representing a ventral view of an adult CNS containing a NB tumor induced during larval stages in the VNC. Green circles are tNBs. Green circles colored in red represent Chinmo$^+$Imp$^+$ tNBs. Immunostainings with anti-Chinmo, anti-Imp and anti-Syp, indicating that Imp/Chinmo and Syp are expressed in a complementary pattern in *pox$^n$ > pros$^{RNAi}$* tumors found in 4-day-old adults. tNBs are labeled with Mira. Scale bar 20 μm. (E) Immunostainings with anti-Chinmo and anti-E93 indicating that these two transcription factors are expressed in a complementary pattern in *pox$^n$ > pros$^{RNAi}$* tumors found in 4-day-old adults. tNBs are marked with anti-Mira. Scale bar 20 μm. (F) The light red and blue colors respectively designate 'early' and 'late' larval NB genes as determined by *Liu et al. (2015)*, *Syed et al. (2017)* and *Ren et al. (2017)*. A subset of these genes are redeployed in tumors to define distinct tNB states. Asterisks mark temporal patterning genes regulated at the post-transcriptional level in tNBs. Note that *cas* and *svp* transcription is associated with early Imp$^+$ NBs, while *br* transcription is associated with late E93$^+$ NBs during larval stages. In contrast, *cas*, *svp* and *br* transcription do not distinguish Imp$^+$ and E93$^+$ tNBs.

DOI: https://doi.org/10.7554/eLife.50375.002

The following figure supplements are available for figure 1:

**Figure supplement 1.** Seurat analysis of the single-cell transcriptomic data, with regression of cell-cycle genes.

DOI: https://doi.org/10.7554/eLife.50375.003

**Figure supplement 2.** Characterizing tNBs by their transcriptome.

DOI: https://doi.org/10.7554/eLife.50375.004

**Figure supplement 3.** Comparison of genes that are temporally regulated genes in various larval NBs with genes defining clusters in the UMAP representation of the NB tumor.

DOI: https://doi.org/10.7554/eLife.50375.005

**Figure supplement 4.** Post-transcriptional regulation of chinmo in *pox$^n$ > pros$^{RNAi}$* NB tumors.

DOI: https://doi.org/10.7554/eLife.50375.006

**Figure supplement 5.** Grh is expressed in all tNBs.

DOI: https://doi.org/10.7554/eLife.50375.007

---

of 5740 cells (22%)) as being strongly defined by the expression of the early NB temporal marker Imp (*Figure 1C* and *Figure 1—figure supplement 2C*). In contrast, E93 expression inversely correlates with Imp expression on the UMAP representation, being strongly expressed throughout all other clusters (*Figure 1C*). The presence of 4 large clusters (clusters 0, 1, 3, 4) within the E93 expression domain on the UMAP representation implies further cellular heterogeneity within the E93$^+$ tNB population (*Figure 1B* and *Figure 1—figure supplement 2C*). We then crossed the list of genes that discriminates the Imp$^+$ cluster 2 from the E93$^+$ clusters (0, 1, 3, 4) with the lists of genes that are temporally regulated in either the mushroom body NBs, antennal lobe NBs or the type II NBs during larval development (*Liu et al., 2015*; *Ren et al., 2017*) (*Figure 1—figure supplement 3*). We found that the genes encoding for the transmembrane transporters Oatp74D and SP1173, the Ig-superfamily transmembrane protein Plum, the glucose metabolism protein Gapdh2, the cytoskeleton protein Chd64, as well as CG10939, CG44325, CG10512 and CG5953 distinguish the Imp$^+$ identity in both NBs during development and tNBs in tumors. In contrast, the CDC25 gene string (stg), CG15628, CG15646 and the long non-coding RNA *noe* constitute a subset of genes whose transcription distinguishes the E93$^+$ identity in both NBs during development and tNBs in tumors (*Figure 1C*, *Figure 1—figure supplement 3* and *Figure 1—figure supplement 2D*). Thus, a subset of larval NB temporal patterning genes is redeployed in tumors defining various tNB states.

Of note, chinmo mRNA is globally strongly expressed throughout all clusters (*Figure 1C*). This is reminiscent of its known post-transcriptional regulation in NBs throughout development (*Dillard et al., 2018*; *Zhu et al., 2006*). Consistent with a post-transcriptional regulation of *chinmo* in tumors, we found that forced transcription of a UAS-mCherry$^{chinmoUTRs}$ transgene, in which the mCherry ORF is flanked by the 5' and 3' UTRs of chinmo (*Dillard et al., 2018*), in the tumor led to mCherry expression only in the tNBs that express the endogenous Chinmo (*Figure 1—figure supplement 4*). Chinmo$^+$ tNBs also co-express Imp (*Figure 1D*). Thus, as in NBs during development, *chinmo* is regulated at the post-transcriptional level in tumors. During development, *chinmo* is post-transcriptionally silenced by Syp whose expression is activated in late NBs (*Liu et al., 2015*; *Ren et al., 2017*; *Syed et al., 2017*). Surprisingly, in tumors, Syp mRNA is not restricted to E93$^+$ tNBs, as it is also highly transcribed in the Imp$^+$ tNBs of cluster 2 (*Figure 1C*). However, when *Syp* expression was assessed by immunostaining in 6 days-old adult tumors, Syp was absent from

Chinmo$^+$Imp$^+$ tNBs. Instead anti-Syp immunostaining labeled most if not all other tNBs, that were also positive for anti-E93 (*Figure 1D,E*). Thus, unlike in larval NBs where *Syp* is transcriptionally controlled along development, its expression in tNBs appears to be regulated mainly at the post-transcriptional level. We also find that the temporal transcription factors *cas* and *svp* that are expressed in Imp$^+$ NBs during early larval stages (*Maurange et al., 2008*; *Ren et al., 2017*; *Syed et al., 2017*) are not enriched in cluster 2 indicating further differences between Imp$^+$ NBs during development and tumorigenesis (*Figure 1—figure supplement 3*). In addition, while the transcription factor Broad (Br) is a pan-NB marker of late temporal identity during larval development (*Liu et al., 2015*; *Maurange et al., 2008*; *Ren et al., 2017*; *Syed et al., 2017*) (*Figure 1—figure supplement 3*), it is not enriched in E93$^+$ tNBs demonstrating differences with late larval NBs (*Figure 1—figure supplement 3*). Lastly, single-cell RNA-seq and immunostainings also showed that all tNBs express the late embryonic and larval temporal transcription factors Grh (*Figure 1—figure supplement 5*) (*Brody and Odenwald, 2000*). This suggests that tNBs do not reverse to an early embryonic temporal identity.

Although *lin-28* is known to be expressed in early larval NBs and in Chinmo$^+$Imp$^+$ tNBs (*Narbonne-Reveau et al., 2016*), mRNA levels appear to stand below the detection limit of the 10x technology, a current limitation of droplet-based assays, as it is hardly detected and not enriched in 'cluster 2' containing Imp$^+$ tNBs (*Figure 1C* and *Figure 1—figure supplement 3*). Low cellular levels of lin-28 mRNA are consistent with a previous RNA-seq experiment on bulk tumors performed by the lab (about 120 times lower than levels of Imp mRNAs) (*Narbonne-Reveau et al., 2016*).

In conclusion, single cell transcriptomics identify a number of larval NB temporal patterning genes that are redeployed in tumors to define distinct tNB subpopulations (*Figure 1F*). Of these, Chinmo$^+$-Imp$^+$ tNBs and Syp$^+$E93$^+$ tNBs, as defined by immunostainings, compose two exclusive subpopulations encompassing most if not all tNBs.

## *pox$^n$ > pros$^{RNAi}$* NB tumors progress to reproducible heterogeneity

To investigate how tumor heterogeneity emerges and is regulated during the course of tumor growth, we co-stained tumors for Chinmo and Syp (respective markers for the Chinmo$^+$Imp$^+$ and Syp$^+$E93$^+$ tNBs) as they grow from their initiation during early larval to adult stages. Immuno-stainings performed about 24 hours after *pros* knockdown (late L2 stage) indicated that pox$^{n+}$ NBs initially amplify to form small pools of Chinmo$^+$Imp$^+$ tNBs (*Figure 2A*).

No Syp$^+$ tNBs are present at this stage. However, from early to late L3, a distinct population of Syp$^+$ tNBs emerges within tumors (*Figure 2B*). Thus, Syp$^+$ tNBs appear in tumors at about the time when normal NBs undergo the Imp-to-Syp switch regulated by the progression of temporal transcription factors (*Maurange et al., 2008*; *Narbonne-Reveau et al., 2016*; *Ren et al., 2017*; *Syed et al., 2017*). In adults, Chinmo$^+$Imp$^+$ tNBs are encountered in small clusters surrounded by large fields of Syp$^+$ tNBs (*Figure 1D*). Thus, the initial pool of Chinmo$^+$Imp$^+$ tNBs evolves to generate tumors with two distinct compartments. Both the Chinmo$^+$Imp$^+$ and Syp$^+$E93$^+$ compartments show mitotic activity (*Figure 2C*), although the mitotic index is globally lower in the Syp$^+$E93$^+$ compartment (*Figure 2D*).

Then, following immunostaining, confocal imaging and 3D reconstruction, we quantified the proportion of Chinmo$^+$Imp$^+$ and Syp$^+$E93$^+$ tNBs within tumors aged of 8, 10, 11 and 15 days (induced in L2 and dissected in 1, 3, 4, and 8-day-old adults respectively). In total, more than 20 tumors were dissected and we reproducibly found a relatively invariant proportion of 20% of Chinmo$^+$Imp$^+$ tNBs (±10) at the different time-points (*Figure 2E*). This proportion is consistent with the proportion of Imp$^+$ tNBs composing 'cluster 2' identified with the single-cell RNA-seq analysis performed from tumors of 4-to-6-day-old adults (tumors induced 11 to 13 days earlier) (*Figure 1C,D*). Thus, Chinmo$^+$Imp$^+$ tNBs globally exhibit a higher mitotic index but, counterintuitively, they rapidly become a minority compared to Syp$^+$E93$^+$ tNBs (*Figure 2E,F*). Note that neurons deriving from tNBs are rare in the bulk of the tumor and only mildly contribute to the tumor mass (*Figure 2—figure supplement 1*). In conclusion, pox$^n$ > pros$^{RNAi}$ tumors rapidly evolve from a homogenous population of Chinmo$^+$Imp$^+$ tNBs in early larvae to a heterogeneous population of Chinmo$^+$Imp$^+$ and Syp$^+$E93$^+$ tNBs in adults. In adults, tNB populations appear to reach a relatively stable equilibrium where Chinmo$^+$Imp$^+$ tNBs are in minority despite an apparent higher mitotic index. The intertumoral reproducibility of this population dynamics suggests the existence of robust intratumoral constraints.

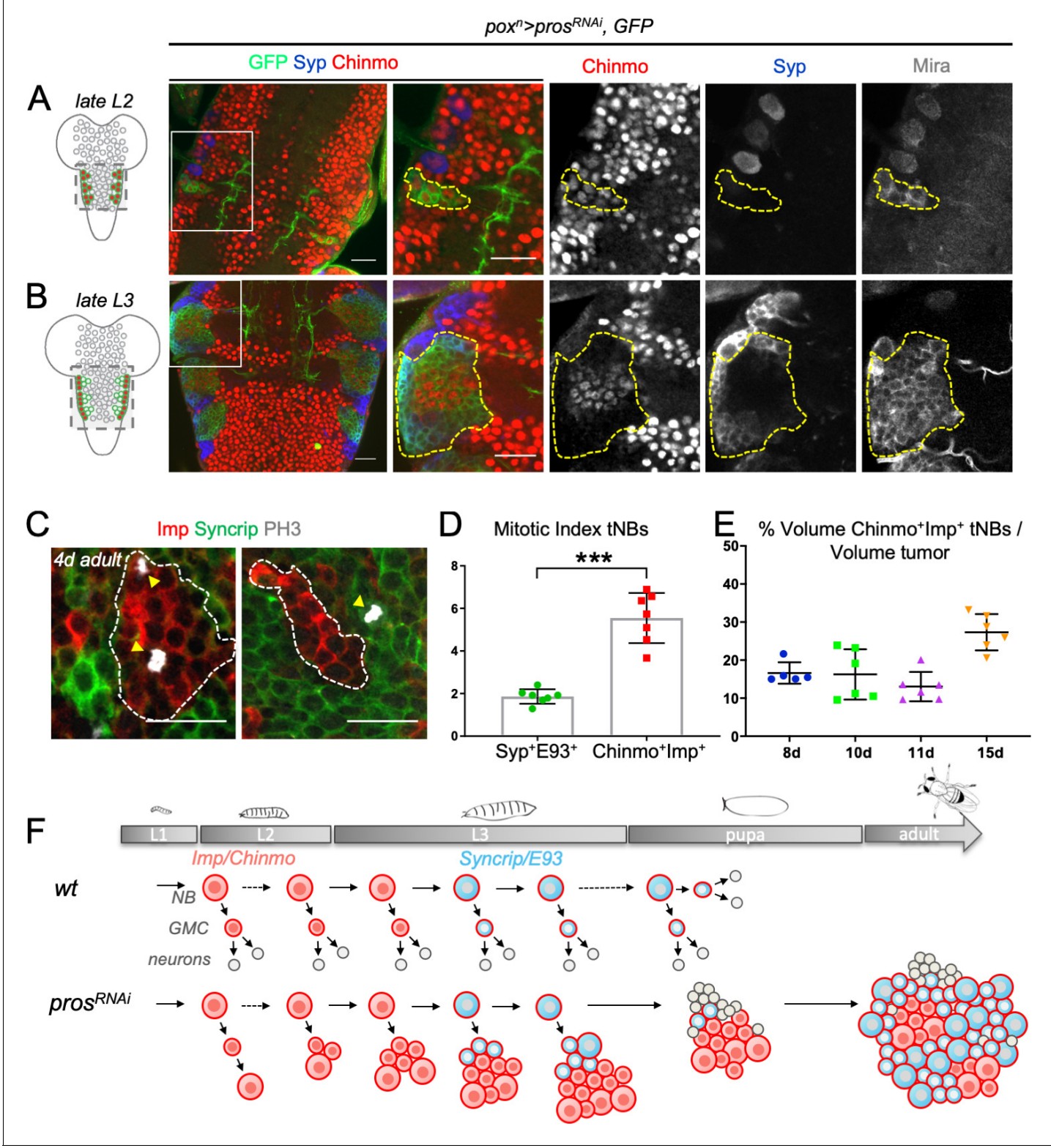

**Figure 2.** Dynamics of cell heterogeneity in $pox^n > pros^{RNAi}$ NB tumors. (**A,B**) Cartoons represent a ventral view of the *Drosophila* CNS at late L2 and late L3. Gray circles represent normal NBs. Green circles are $pros^{RNAi}$ tNBs. Red tNBs express *chinmo*. Tumors were induced by knocking down *pros* in six NBs located in the VNC, throughout larval development, using the *pox^n-Gal4, UAS-pros^{RNAi}, UAS-GFP, UAS-dicer2* system ($pox^n > pros^{RNAi}$). Because *chinmo* is always co-expressed with *Imp* in NBs, we use either anti-Chinmo or anti-Imp to label Chinmo+Imp+ tNBs. We also used Syp to label Syp+E93+ tNBs. (**A**) $pox^n > pros^{RNAi}$ initially induces pools of tNBs all expressing Chinmo in early larvae (L2). tNBs are marked with anti-Mira and anti-
*Figure 2 continued on next page*

 Research advance                                      Cancer Biology | Developmental Biology

*Figure 2 continued*

GFP. (**B**) In late larvae (L3), $pox^n > pros^{RNAi}$ tumors are composed by two distinct populations of tNBs respectively expressing Chinmo and Syp. Tumors are marked with anti-GFP and delineated by dashed lines. (**C**) Mitotic Imp$^+$ tNBs and Syp$^+$tNBs are marked with anti-PH3 in $pox^n > pros^{RNAi}$ tumors persisting in 4-day-old adults. (**D**) Quantification of the mitotic index of Chinmo$^+$Imp$^+$ tNBs (n = 7 VNCs) and Syp$^+$E93$^+$ tNBs (n = 7 VNCs) in $pox^n > pros^{RNAi}$ tumors of 4-day-old adults. p=0.0006. (**E**) Proportion of Chinmo$^+$Imp$^+$ tNBs over all tNBs composing tumors (volumes of each population are measured) at 8 days (8d) (n = 5), 10d (n = 6), 11d (n = 6) and 15d (n = 6) after tumor induction. Each dot represents the % for one tumor. (**F**) Scheme depicting the dynamics of tumor composition: from a homogeneous pool of Chinmo$^+$Imp$^+$ tNBs in early larvae to a heterogeneous tumor with a minor population of Chinmo$^+$Imp$^+$ and a majority of Syp$^+$E93$^+$ tNBs. Scale bars, 20 μm.
DOI: https://doi.org/10.7554/eLife.50375.008

The following figure supplement is available for figure 2:

**Figure supplement 1.** $pox^n > pros^{RNAi}$ tumors exhibit low levels of neuronal differentiation in adults.
DOI: https://doi.org/10.7554/eLife.50375.009

## $pox^n > pros^{RNAi}$ NB tumors follow a rigid hierarchical scheme

To investigate in details the rules governing the population dynamics of Chinmo$^+$Imp$^+$ and Syp$^+$E93$^+$ tNBs, we designed lineage analysis experiments in vivo. We used the Flybow technique combined with our $pox^n > pros^{RNAi}$ system to generate random mCherry-labeled clones in otherwise GFP$^+$ tumors (*Hadjieconomou et al., 2011*). Clones were randomly induced at low frequency (in about 1% to 2% of tNBs) in 2-day-old adults containing $pox^n > pros^{RNAi}$ tumors, and examined along a time course. Of note, even 12 days after clonal induction (ACI), clones could be easily delineated, with mCherry$^+$ cells depicted no or little dispersion denoting no active migration (*Figure 3—figure supplement 1*). This allowed us to characterize individual mCherry$^+$ clones 8 hours, 2 days, 4 days and 8 days ACI (*Figure 3A*). We used Chinmo as a marker for Chinmo$^+$Imp$^+$ tNBs and its absence as a marker for Syp$^+$E93$^+$ tNBs. Along this time lapse, we could observe three categories of clones: clones exclusively composed of Chinmo$^+$Imp$^+$ tNBs (further termed CI clones), MIXED clones composed of both Chinmo$^+$Imp$^+$ and Syp$^+$E93$^+$ tNBs (they therefore contain at least two tNBs), and clones exclusively composed of Syp$^+$E93$^+$ tNBs (further termed SYP clones) (*Figure 3C*). The existence of clones exclusively composed of either Chinmo$^+$Imp$^+$ or Syp$^+$E93$^+$ tNBs is consistent with the ability of both types of tNBs to undergo self-renewing divisions. In addition, the existence of a subset of MIXED clones implies that Chinmo$^+$Imp$^+$ or Syp$^+$E93$^+$ tNBs may derive from a common precursor of either identity. In addition, clones display a large heterogeneity in their size at 8 days ACI (*Figure 3A,B*), indicating that all tumor cells do not possess the same proliferation potential. Interestingly, we observed that the proportion of the different categories of clones is dynamic over the period of 8 days. The proportion of CI clones exhibits a rapid decrease that is paralleled with an increase in the proportion of MIXED clones and a slight increase in the proportion of SYP clones (*Figure 3D*). We sought to investigate whether these clonal dynamics could reflect hierarchical rules within the tumor.

For this purpose, we designed a stochastic numerical model of clone growth (see Materials and methods for details). The model follows a simple algorithm (*Figure 4A*). Each clone starts from a single tNB. In the model, Chinmo$^+$Imp$^+$ tNBs are referred to as C cells while Syp$^+$E93$^+$ tNBs are referred to as S cells. The initial tNB can either be Chinmo$^+$Imp$^+$ (probability $p_c$) or Syp$^+$E93$^+$ (probability $1 - p_c$). At each numerical time step, each cell in the clone (initially one) has a given probability to divide, set by its division time ($T_c$ and $T_s$). Upon division, Chinmo$^+$Imp$^+$ tNBs can either duplicate (C→CC), generate two Syp$^+$E93$^+$ tNBs (C→SS), or undergo fate asymmetric division (C→CS). Similarly, Syp$^+$E93$^+$ tNBs can either duplicate (S→SS), generate two Chinmo$^+$Imp$^+$ tNBs (S→CC), or undergo fate asymmetric division (S→CS). Each new tNB has a probability to exit the cell-cycle and enter quiescence. In modeling terms, we asked whether hierarchical or plastic schemes of cell divisions are compatible with the proportions of clone categories observed experimentally.

We first tested four simplified scenarios, in which we neglected quiescence and asymmetric divisions, and assumed that C and S cells have the same division time. In each case, we simulated the growth of 1000 clones and plotted the proportion of clones in each category. In the first scenario ('No hierarchy'), Chinmo$^+$Imp$^+$ tNBs have 50% chance to duplicate (symmetric self-renewing divisions C→CC), and 50% chance to divide into two Syp$^+$E93$^+$ tNBs (C→SS). Similarly, Syp$^+$E93$^+$ tNBs have 50% chance to duplicate (S→SS), and 50% chance to divide into two Chinmo$^+$Imp$^+$ tNBs (S→CC). Unlike our experimental observations, this non-hierarchical scenario leads to a symmetric

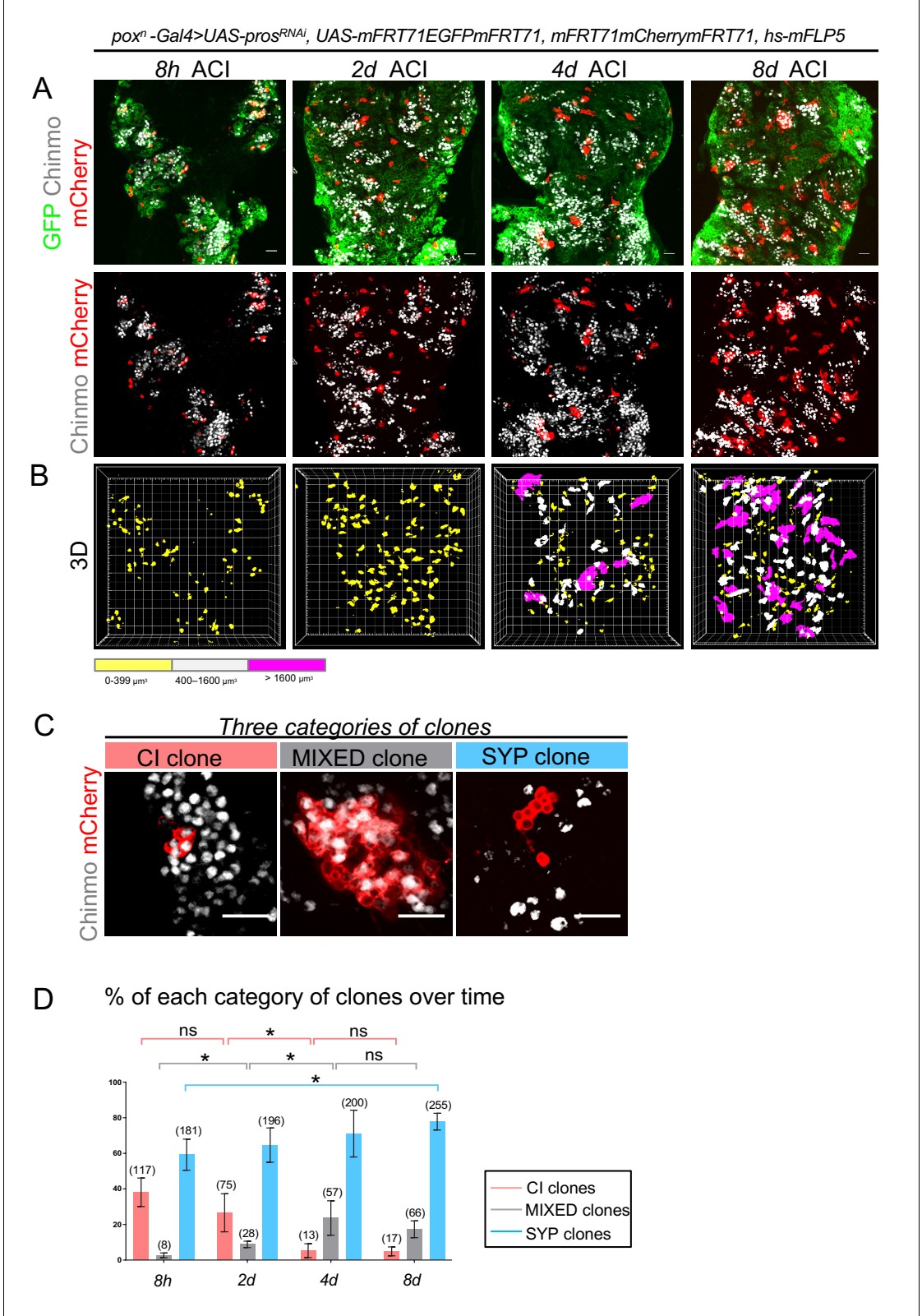

**Figure 3.** Clonal analysis in $pox^n > pros^{RNAi}$ NB tumors. (**A**) Clones are labeled with mCherry and observed 8 hours (8h), 2 days (2d), 4 days (4d) and 8 days (8d) after clonal induction (ACI) in $pox^n > pros^{RNAi}$ tumors. mCherry⁻ tNBs are GFP⁺. Images represent one confocal section. (**B**) 3D projections of clones 8 hr, 2d, 4d and 8d ACI. The color-code labels clones according to their volume. (**C**) Three categories of clones can be identified in $pox^n > pros^{RNAi}$ tumors: clones composed of Chinmo⁺Imp⁺ tNBs only (CI clones), clones composed of both Chinmo⁺Imp⁺ and Syp⁺E93⁺ tNBs (MIXED

*Figure 3 continued on next page*

*Figure 3 continued*

clones), and clones composed of Syp⁺E93⁺ tNBs only (SYP clones). Chinmo⁺Imp⁺ tNBs are identified by the presence of Chinmo. Syp⁺ tNBs are identified by the absence of Chinmo. (D) Proportion of CI (red), MIXED (gray) and SYP (blue) clones 8h, 2d, 4d and 8d ACI. Proportion of CI clones 8h ACI (n = 117 clones from 4 VNCs); 2d ACI (n = 75 clones from 4 VNCs); 4d ACI (n = 13 clones from 5 VNCs); 8d ACI (n = 17 clones from 5 VNCs). $P$ between CI clones at 8h and 2d ACI -> $P_{CI8h/2d} = 0.2$; $P_{CI2d/4d} = 0.016$; $P_{CI4d/8d} = 0.88$. Proportion of MIXED clones 8h ACI (n = 8 clones from 4 VNCs); 2d ACI (n = 28 clones from 4 VNCs); 4d ACI (n = 57 clones from 5 VNCs); 8d ACI (n = 66 clones from 5 VNCs). $P_{MIXED8h/2d} = 0.029$; $P_{MIXED2d/4d} = 0.016$; $P_{MIXED4d/8d} = 0.31$. Proportion of SYP clones 8h ACI (n = 181 clones from 4 VNCs); 2d ACI (n = 196 clones from 4 VNCs); 4d ACI (n = 200 clones from 5 VNCs); 8d ACI (n = 255 clones from 5 VNCs). $P_{SYP8h/8d} = 0.016$. Scale bars, 20 μm.

DOI: https://doi.org/10.7554/eLife.50375.010

The following figure supplement is available for figure 3:

**Figure supplement 1.** tNBs within a clone do not disperse.

DOI: https://doi.org/10.7554/eLife.50375.011

outcome, in which all clones rapidly become mixed, while the proportion of CI and SYP clones rapidly decays to zero (*Figure 4B*). In the second scenario ('Strict hierarchy #1'), Chinmo⁺Imp⁺ tNBs still have 50% chance to duplicate (C→CC), and 50% chance to divide into two Syp⁺E93⁺ tNBs (C→SS) but Syp⁺E93⁺ tNBs can only self-renew (S→SS). This hierarchical scheme leads to a fully different outcome, with a large proportion of clones remaining only composed of Syp⁺E93⁺ tNBs (SYP clones), while the proportion of clones only composed of Chinmo⁺Imp⁺ tNBs (CI clones) rapidly decays, which is consistent with our experimental observations (*Figure 4B*). In the third scenario, we also tested the reverse hierarchical scheme ('Strict hierarchy #2') where Syp⁺E93⁺ tNBs still have 50% chance to duplicate (S→SS), and 50% chance to divide into two Chinmo⁺Imp⁺ tNBs (S→CC) but Chinmo⁺Imp⁺ tNBs can only self-renew (C→CC). Consistently, we observed results opposite to the second scenario, with CI clones rapidly becoming dominant instead of SYP clones (*Figure 4B*). These scenarios therefore suggest that $pox^n > pros^{RNAi}$ NB tumors follow a rather hierarchical scheme with Chinmo⁺Imp⁺ tNBs at the top of the hierarchy. Finally, we tested a fourth 'Plastic hierarchy' scenario for which we maintain this hierarchy, but also give Syp⁺E93⁺ tNBs a small probability (1%) to generate Chinmo⁺Imp⁺ tNBs (S→CC). This is sufficient to drastically change the outcome of the simulations, as it prevents the long-term maintenance of SYP clones (*Figure 4B*, right panel). These four model scenarios combined to our experimental observations therefore argue for a rigid, unidirectional, Chinmo⁺Imp⁺→Syp⁺E93⁺ hierarchy between tNBs.

## The hierarchical division scheme that defines clonal growth and composition demonstrates a cancer stem cell-like role for Chinmo⁺Imp⁺ tNBs

We then sought to use the hierarchical model #1 as a basis to investigate more quantitatively the parameters that finetune tumor growth and heterogeneity. Interestingly, in vivo, significant differences in average clone size can be observed between the three categories at the different time points (*Figure 4C*). SYP clones grow slower than MIXED clones, suggesting either that Syp⁺E93⁺ tNBs have a slower cell-cycle speed or possess a limited self-renewing potential - similar to Syp⁺E93⁺ NBs during development – and rapidly end up exiting the cell cycle. On the other hand, the few clones that remain within the CI category at 8 days ACI tend to completely stop growing, suggesting that these clones may be composed by quiescent Chinmo⁺Imp⁺ tNBs. Consequently, we considered non-zero probabilities $q_s$ and $q_c$ for new S cells and C cells to enter quiescence. We also implemented a non-zero probability for C cells to undergo fate asymmetric divisions (*Figure 4—figure supplement 1*), and distinct division times for C and S cells. Thanks to a series of measurements, we could reduce the number of independent parameters to two (see Materials and methods): the probability $P(c \rightarrow cc)$, and the division time $T_s$ for S cells. We used these as free parameters to fit the model to the measurements of clone sizes and compositions at 8 hours, 2, 4 and 8 days ACI. To that end, we computed error maps for clone composition and clone size in the $(Pc \rightarrow cc, \ T_s)$ plane, and eventually a combined error map (*Figure 4—figure supplement 2*). We found that the error is minimized for $P(c \rightarrow cc) = 0.64$ and $T_s = 1.3$ days, from which we determine all the other parameters. Notably, Syp⁺ tNBs have a much higher probability (47%) to enter quiescence than Chinmo⁺Imp⁺ tNBs (4%), consistent with the slower growth of SYP clones. Using the set of parameters determined by the fit (*Figure 4F*), we found an excellent agreement between experiments and simulations, for both clone

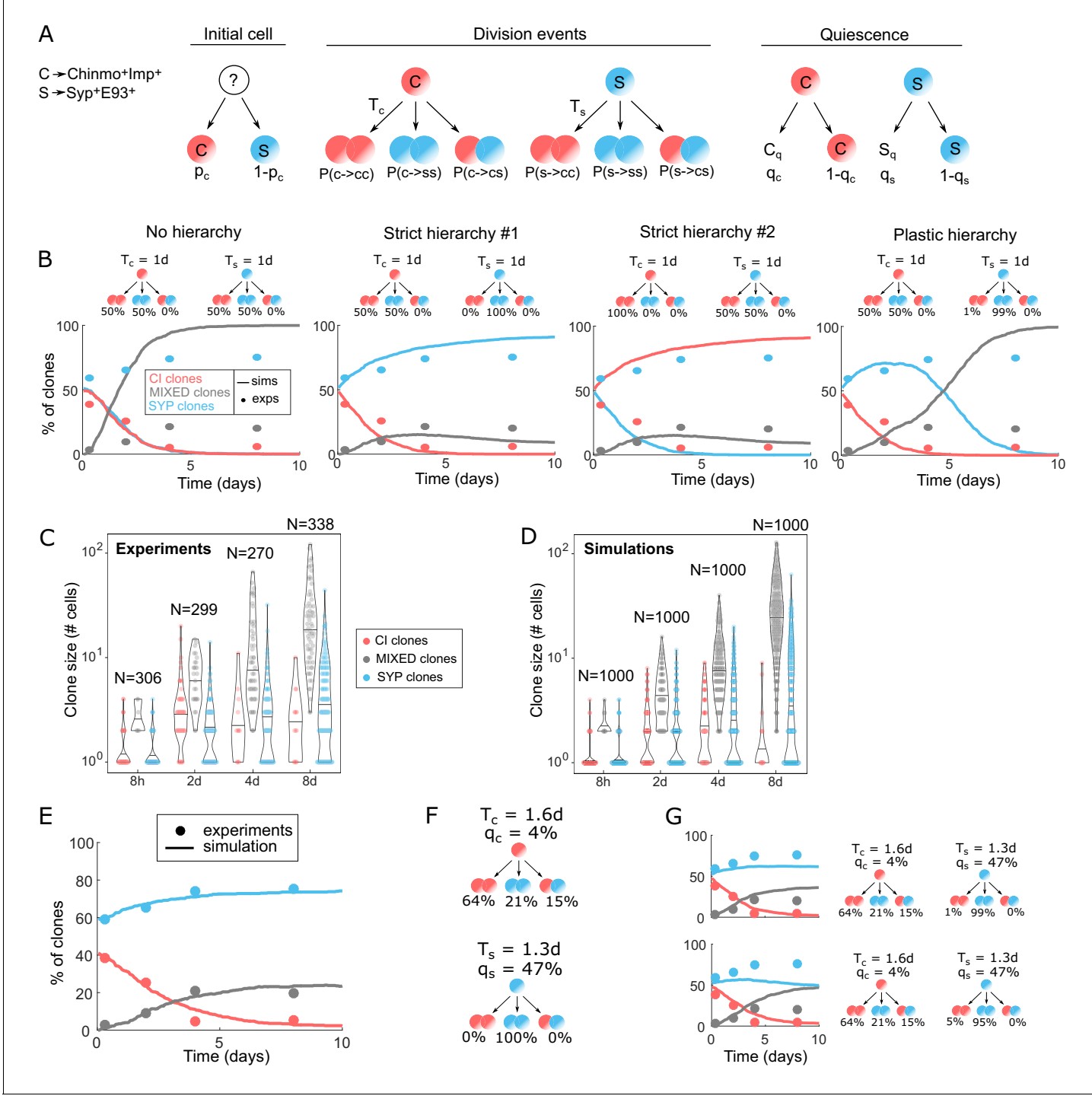

**Figure 4.** Modeling tumor hierarchy and the dynamics of cellular heterogeneity. (A) Cartoon of the stochastic clone model. C stands for Chinmo⁺Imp⁺ tNBs, S stands for Syp⁺E93⁺ tNBs. $T_c$ means 'division time of Chinmo⁺Imp⁺ tNBs', $T_s$ means 'division time of Syp⁺E93⁺ tNBs'. $C_q$ stands for quiescent Chinmo⁺Imp⁺ tNB, $S_q$ stands for quiescent Syp⁺E93⁺ tNB. $q_c$ and $q_s$ are the probabilities for Chinmo⁺Imp⁺ and Syp⁺E93⁺ tNBs to be quiescent after being generated. (B) Dynamics of proportion for each clone category in four extreme model scenarios: No hierarchy, Strict hierarchy #1, Strict hierarchy #2, Plastic hierarchy. Solid lines represent the simulations (n = 1000 clones), dots represent experimental measurements. Cartoons above graphs represent the division probabilities used to generate each graph. (C) Violin plots depicting distributions of clone sizes (number of cells) for each category calculated from the experiments. (D) Violin plots depicting distributions of clone sizes (number of cells) for each category of clones after simulation, using the set of parameters minimizing the error. Center lines of boxes show the medians. (E) Proportion of clones over time in each category using the set of parameters minimizing the error. Solid lines represent the simulations (n = 1000 clones), dots represent experimental

*Figure 4 continued on next page*

*Figure 4 continued*

measurements. (F) Hierarchical scheme able to recapitulate the dynamics of clone growth and composition, with the parameters measured from the experimental data and defined by the fit. (G) Proportion of clones in each category using the set of parameters minimizing the error while allowing a small chance of reverse division from S to C (top: 1%, bottom: 5%).

DOI: https://doi.org/10.7554/eLife.50375.012

The following figure supplements are available for figure 4:

**Figure supplement 1.** Three categories of two-cell clones in $pox^n > pros^{RNAi}$ NB tumors suggesting that tNBs can undergo fate symmetric and fate asymmetric divisions .

DOI: https://doi.org/10.7554/eLife.50375.013

**Figure supplement 2.** Error maps Normalized error maps in the (Ts,P(c->cc)) space.

DOI: https://doi.org/10.7554/eLife.50375.014

sizes (*Figure 4C, D*) and clone compositions (*Figure 4E*). Modulating these parameters to allow a small probability for S→CC reverse divisions (1% or 5%) significantly affected the simulations outcome, leading to an important loss of SYP clones and a concomitant increase of MIXED clones (*Figure 4G*). This further demonstrates that reverse divisions from $Syp^+E93^+$ tNBs to $Chinmo^+Imp^+$ tNBs are either very low or inexistent, at least during the first weeks of tumor growth.

Altogether, our clonal analysis demonstrates that NB tumors are strongly hierarchical. $Chinmo^+$-$Imp^+$ tNBs are at the top at the hierarchy. They have a preference for symmetric self-renewing divisions, allowing their perpetuation, and rarely become quiescent allowing tumor growth propagation. They can also undergo fate asymmetric or symmetric differentiation divisions in order to generate $Syp^+E93^+$ tNBs, albeit with a lower probability. Subsequent $Syp^+E93^+$ tNBs can undergo symmetric self-renewing divisions but exhibit a high propensity for rapid cell-cycle exit and cannot generate $Chinmo^+Imp^+$ tNBs. Consequently, $Syp^+E93^+$ tNBs cannot generate large and heterogeneous clones in tumors unlike $Chinmo^+Imp^+$ tNBs. This set of characteristics (*Figure 4F*) thus confers cancer stem cell (CSC)-like properties to $Chinmo^+Imp^+$ tNBs (*Nassar and Blanpain, 2016*; *Nguyen et al., 2012*; *Valent et al., 2012*).

## Tumor composition is predictable but varies according to the cell of origin

We then tested whether the set of parameters defined by our analysis of clonal growth could recapitulate the tNB population dynamics, from homogeneity (in early larvae) to stable heterogeneity (in adults). When applying the uncovered set of parameters to a homogenous pool of $Chimo^+Imp^+$ tNBs as observed in early larvae (*Figure 2A*), simulations via the stochastic or a deterministic version of the model (see Materials and methods) predict that $Chimo^+Imp^+$ and $Syp^+E93^+$ tNB populations rapidly reach an equilibrium with a 20/80 ratio (*Figure 5A*). Remarkably, this is consistent with our observations that the population of $Chinmo^+Imp^+$ tNBs invariably progresses to the minority, and reaches a similar equilibrium in vivo (*Figure 2E*). Our observation that all $pox^n > pros^{RNAi}$ tumors invariably reach an approximate 20/80 equilibrium during adulthood suggests a robust hierarchical scheme able to buffer intrinsic or extrinsic perturbations inherent to biological systems. Interestingly, changing the initial proportions of tNBs in the simulation does not affect the final equilibrium (*Figure 5B*). Thus, the uncovered hierarchical scheme robustly reproduces the observed dynamics of intratumoral heterogeneity, at least during the first two and half weeks of tumor growth (from L1 to 10-day-old adults) and provides robustness to accommodate biological perturbations.

To investigate whether the 20/80 ratio is a property of all NB tumors, we measured this ratio in tumors induced by the loss of the SWI/SNF factor *snr1* in larval type II NBs of the central brain (*Eroglu et al., 2014*; *Koe et al., 2014*). As for $pox^n > pros^{RNAi}$ tumors, late larval and adult NB tumors induced by a null *Snr1* allele or $Snr1^{RNAi}$ are composed by a mix of $Chinmo^+Imp^+$ tNBs and $Syp^+$ tNBs (*Figure 5C*). As for $pox^n > pros^{RNAi}$ tumors the ratio of $Chinmo^+Imp^+$ over $Syp^+$ tNBs is reproducible from adult to adult tumors (*Figure 5D*). However, we observed that it is different from $pox^n > pros^{RNAi}$ tumors with an approximate 50/50 ± 10 ratio encountered in $Snr1^{RNAi}$ tumors. Thus, the rules finetuning the parameters defining the hierarchical scheme in the different types of tumors are robust, but tumor-specific, possibly determined by the NB of origin.

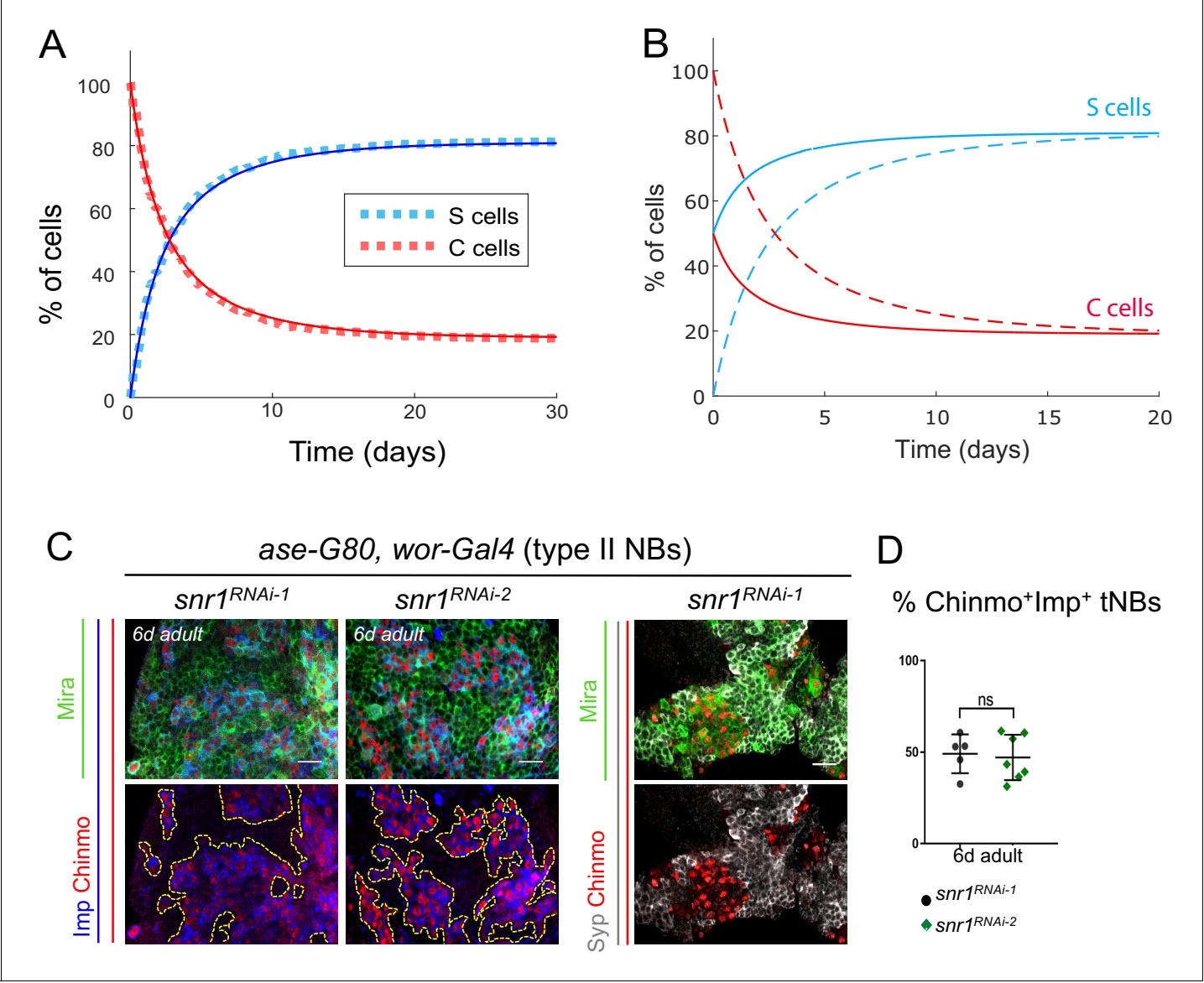

**Figure 5.** The final proportions of Chinmo⁺Imp⁺ and Syp⁺E93⁺ tNBs are not influenced by their initial proportions and depend on the tumor type. (**A**) Proportion of Syp⁺E93⁺ (blue) and Chinmo⁺Imp⁺ (red) tNBs over time using the set of parameter values determined by the error minimization and starting from 100% of Chinmo⁺Imp⁺ tNBs. Dashed lines represent a stochastic simulation (clone model), thin lines represent the prediction of the deterministic model. An equilibrium is rapidly reached were Chinmo⁺Imp⁺ tNBs represent about 20% of all tNBs. (**B**) Dashed curves: evolution of the proportions of Syp⁺E93⁺ (blue) and Chinmo⁺Imp⁺ (red) tNBs, as predicted by the deterministic model, when starting with 100% of Chinmo⁺Imp⁺. Plain curves: evolution of the proportions of Syp⁺E93⁺ (blue) and Chinmo⁺Imp⁺ (red) tNBs, as predicted by the deterministic model, when starting with 50% of Chinmo⁺Imp⁺ and 50% of Syp⁺E93⁺ tNBs. In both scenarios, the same equilibrium is reached. (**C**) *UAS-snr1*^RNAi^ transgenes are expressed in type II NBs using the *ase-Gal80, wor-Gal4, UAS-dicer2* system causing NB tumors that persist in the CB of adult flies. tNBs in 6-day-old adults are marked with Mira, Imp, Chinmo and Syp. (**D**) Proportion of Chinmo⁺Imp⁺ tNBs in *ase-G80, wor > snrRNAi*^RNAi-1^ (n = 5 CB) and *ase-G80, wor > snrRNAi*^RNAi-2^ (n = 7 CB) tumors in 6-day-old adults (volume of Chinmo+Imp+ tNBs over volume of all tNBs x 100) . p=0.0061. Scale bars, 20 μm.
DOI: https://doi.org/10.7554/eLife.50375.015

## Imp and Syp antagonistically regulate NB growth and self-renewal during development and tumorigenesis via *chinmo*

We then investigated the role of the two RNA-binding proteins Imp and Syp in tumor growth and cellular heterogeneity. We had previously demonstrated that Imp sustains NB tumor growth beyond normal developmental stages at least partially by promoting *chinmo* expression (**Narbonne-**

*Reveau et al., 2016*). To investigate the function of Syp within the tumor context, we knocked it down in $pox^n > pros^{RNAi}$ tumors from their initiation. *Syp* knockdown led to tumors almost exclusively constituted of Chinmo[+]Imp[+] tNBs (*Figure 6A,B*). This is consistent with the finding that *Syp* knockdown induces maintenance of both Imp (*Yang et al., 2017*) and Chinmo in NBs during

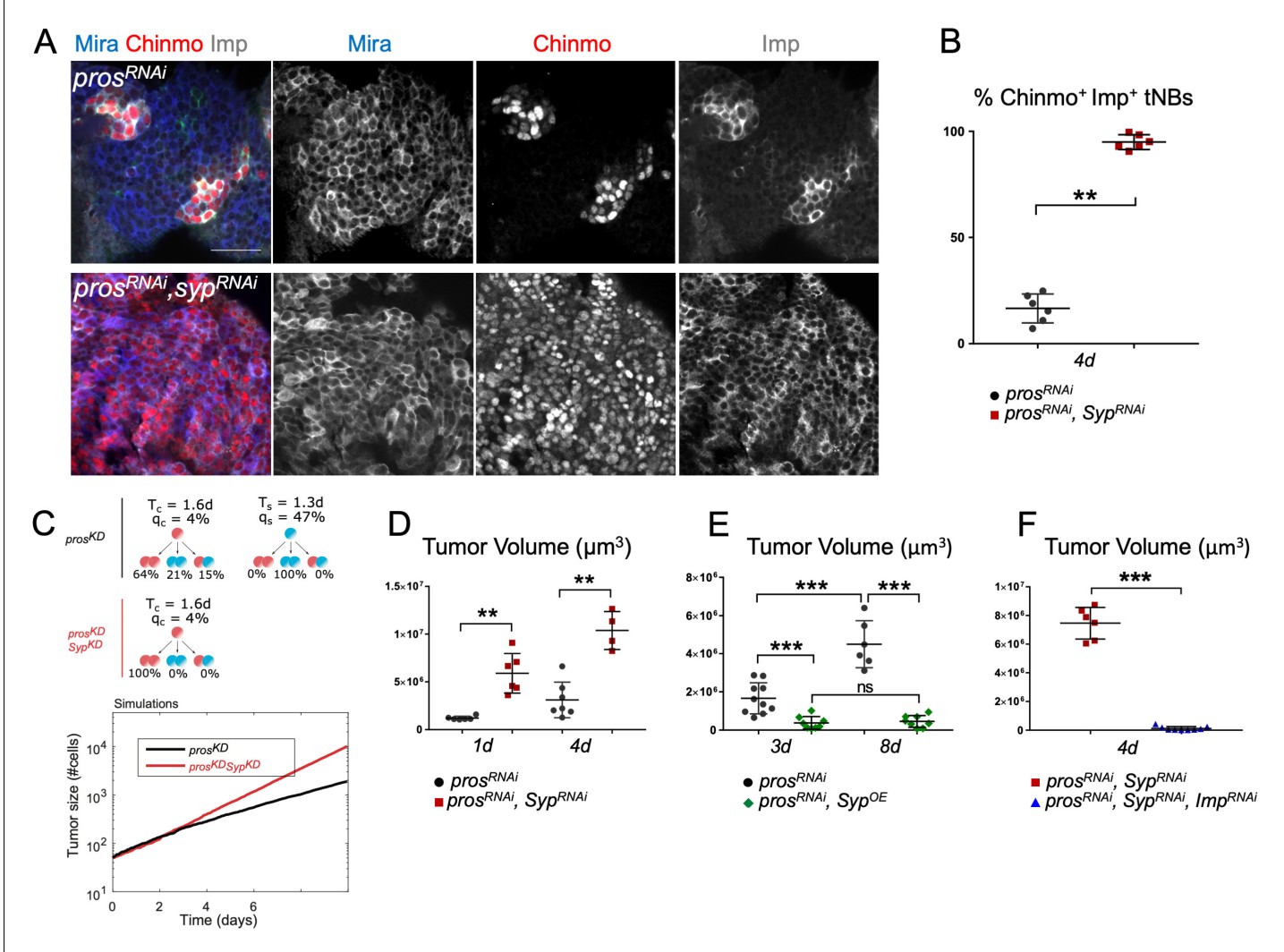

**Figure 6.** Imp and Syp antagonistically regulate the growth and heterogeneity of $pros^{RNAi}$ tumors. (A) Syp knockdown in $pox^n > pros^{RNAi}$ tumors triggers a dramatic increase of Chinmo[+]Imp[+] tNBs. tNBs are marked with an anti-Mira antibody. (B) Proportion of Chinmo[+]Imp[+] tNBs in $pox^n > pros^{RNAi}$ (n = 6 VNC) and $pox^n > pros^{RNAi}, Syp^{RNAi}$ (n = 6 VNC) tumors in 4-day-old adults. p=0.0022. (C) Simulation of tumor growth (number of cells) in $pros^{KD}$ and $pros^{KD}, Syp^{KD}$ (red) tumors. Quiescence probabilities are those set by the error minimization. (D) Volume of $pox^n > pros^{RNAi}$ (n = 6 VNC) and $pox^n > pros^{RNAi}, Syp^{RNAi}$ (n = 6 VNC) tumors in 1-day-old adults. p=0.002. Volume of $pox^n > pros^{RNAi}$ (n = 7 VNC) and $pox^n > pros^{RNAi}, Syp^{RNAi}$ (n = 4 VNC) tumors in 4-day-old adults. p=0.0061. (E) Volume of $pox^n > pros^{RNAi}$ tumors in 3-day-old (n = 10 VNC) and 8-day-old (n = 6 VNC) adults. p=0.00025. Volume of $pox^n > pros^{RNAi}$ (n = 10 VNC) and $pox^n > pros^{RNAi}, UAS\text{-}Syp$ (n = 8 VNC) tumors in 3-day-old adults. p=0.0005. Volume of $pox^n > pros^{RNAi}$ (n = 6 VNC) and $pox^n > pros^{RNAi}, UAS\text{-}Syp$ (n = 8 VNC) tumors in 8-day-old adults. p=0.00067. Volume of $pox^n > pros^{RNAi}, Syp^{OE}$ tumors in 3-day-old (n = 8 VNC) and 8-day-old (n = 8 VNC) adults. p=0.7209. (F) Volume of $pox^n > pros^{RNAi}, Syp^{RNAi}$ (n = 6 VNC) and $pox^n > pros^{RNAi}, Imp^{RNAi}, Syp^{RNAi}$ (n = 9 VNC) tumors in 4-day-old adults. p=0.0004.

DOI: https://doi.org/10.7554/eLife.50375.016

The following figure supplements are available for figure 6:

**Figure supplement 1.** Syp silences Chinmo and Imp to limit NB self-renewal during development.
DOI: https://doi.org/10.7554/eLife.50375.017

**Figure supplement 2.** Imp and Syp antagonistically regulate tumor growth and *chinmo* expression in $pox^n > pros^{RNAi}$ NB tumors.
DOI: https://doi.org/10.7554/eLife.50375.018

development (*Figure 6—figure supplement 1*). This suggests that in absence of Syp, Chinmo$^+$Imp$^+$ tNBs tend to exclusively undergo symmetric self-renewing divisions. As predicted by the numerical model, this should lead to an increased tumor growth rate, since Chinmo$^+$Imp$^+$ tNBs do not, or rarely enter quiescence, unlike Syp$^+$E93$^+$ tNBs (*Figure 6C*). Consistently, in vivo, the volume of *pros$^{RNAi}$*; *Syp$^{RNAi}$* tumors in 1-day-old adults exhibited a 3.9-fold increase compared to control *pros$^{RNAi}$* tumors (*Figure 6D* and *Figure 6—figure supplement 2A*). By contrast, over-expressing *Syp* (*Syp$^{OE}$*) within *pros$^{RNAi}$* tumors blocked tumor growth in adults (*Figure 6E* and *Figure 6—figure supplement 2A*). In addition, knockdown of both *Syp* and *Imp* in *pros$^{RNAi}$*, *Syp$^{RNAi}$*, *Imp$^{RNAi}$* tumors arrested tumor growth in adults (*Figure 6F* and *Figure 6—figure supplement 2*), and tumors lacked or exhibited low levels of Chinmo (*Figure 6—figure supplement 2*) demonstrating that, even in the absence of Syp, Imp is essential for Chinmo expression in tumors and that is required for their continuous growth. Thus, the overgrowth phenotype observed in *pros$^{RNAi}$*, *Syp$^{RNAi}$* tumors is mediated by the Chinmo/Imp module. Together, these experiments indicate that cellular heterogeneity is not required for tumor growth, and that Syp acts as a tumor suppressor. Syp restrains the population of Chinmo$^+$Imp$^+$ CSC-like tNBs and its inactivation in tumors abolishes the hierarchy.

We then tested whether Imp and Syp exert their antagonistic activities by competing on the same RNAs. As revealed by affinity pull-down assays performed using in vitro synthesized chinmo 5' and 3'UTRs, both Imp and Syp can associate with the UTRs of chinmo mRNA (*Figure 6—figure supplement 1A*). Therefore, Imp and Syp can have the same target genes. Interestingly, Syp appears to have a greater affinity for the 5'UTR (*Figure 6—figure supplement 1A*). This phenomenon provides a molecular explanation for the observation that chinmo 5'UTR is required for post-transcriptional silencing (*Zhu et al., 2006*). Together, our data indicate that NB tumor growth, hierarchy and cellular heterogeneity is governed by the finetuned activities of the two RNA-binding proteins Imp and Syp, largely via the post-transcriptional regulation of *chinmo*.

## tNBs follow a trajectory partially recapitulating larval temporal patterning

We then thought to use our single-cell RNA-seq data to investigate in more details the genes expressed along the Imp/Syp-mediated tumor hierarchy. For this purpose, we used Monocle 2 to perform a pseudotime analysis (*Qiu et al., 2017a*; *Qiu et al., 2017b*; *Trapnell et al., 2014*). Pseudotime analysis determines trajectories of differentiation by ordering cells according to their transcriptional changes. Our numerical model demonstrated that Chinmo$^+$Imp$^+$ tNBs are at the top of the tumor hierarchy. Therefore, tNBs with high levels of Imp can be positioned at the root of the pseudotime trajectory. In contrast, Syp$^+$E93$^+$ tNBs lie further down in the hierarchy and E93 marks late NBs during development (*Ren et al., 2017*; *Syed et al., 2017*). In consequence, tNBs expressing high levels of E93 can be assigned to the end of a differentiation trajectory. Based on these assumptions, we ran a semi-supervised ordering to uncover genes that are dynamically expressed along the Imp→E93 differentiation trajectory that occurs within *pox$^n$ > pros$^{RNAi}$* tumors. The pseudotime ordering generated a trajectory that is initially enriched in Imp$^+$ tNBs and that splits into two main branches enriched in E93$^+$ tNBs (*Figure 7A,B*). We then searched for the top 200 genes whose expression changes as a function of pseudotime. They could be separated in three main clusters (*Figure 7C* and *Supplementary file 1*): Cluster 1 contains genes that, similar to Imp, decrease their expression along the pseudotime (111 genes). They include *Oatp74D, Sip1/CG10939, plum/ CG6490, Chd64, SP1173, CG10512, Gapdh2* and CG44325 (*Figure 7B*) that are expressed in early larval NBs before the Imp-to-Syp switch (*Figure 1F*). Of note *cas* and *svp*, the upstream temporal transcription factors scheduling the Imp-to-Syp transition in larval NBs are absent from cluster 1 consistent with our observation that they are not enriched in Imp$^+$ tNBs (*Figure 1—figure supplement 3*). Cluster 2 and 3 contains genes that, similar to E93, increase their expression along the pseudotime (71 and 18 genes respectively). However, they discriminate genes that are differentially expressed along the two branches of the differentiation trajectories (*Figure 7A*). Cluster 2 is enriched in ribosomal proteins, and also includes the late larval NB temporal markers: *stg, lncRNA: noe,* and *CG15628* (*Figure 7C,D* and *Supplementary file 1*). In the tumor, *stg* is enriched in the branch defined by state three along the trajectory (*Figure 7A*). In contrast, cluster 3 pinpoints genes that are highly expressed in the branch defined by state 7 (*Figure 7A*) including the late temporal patterning gene CG15646. Interestingly, cluster 3 is also strongly enriched in genes of the E(spl) family, known targets of the Notch pathway (*Figure 7B* and *Supplementary file 1*). Thus, the expression

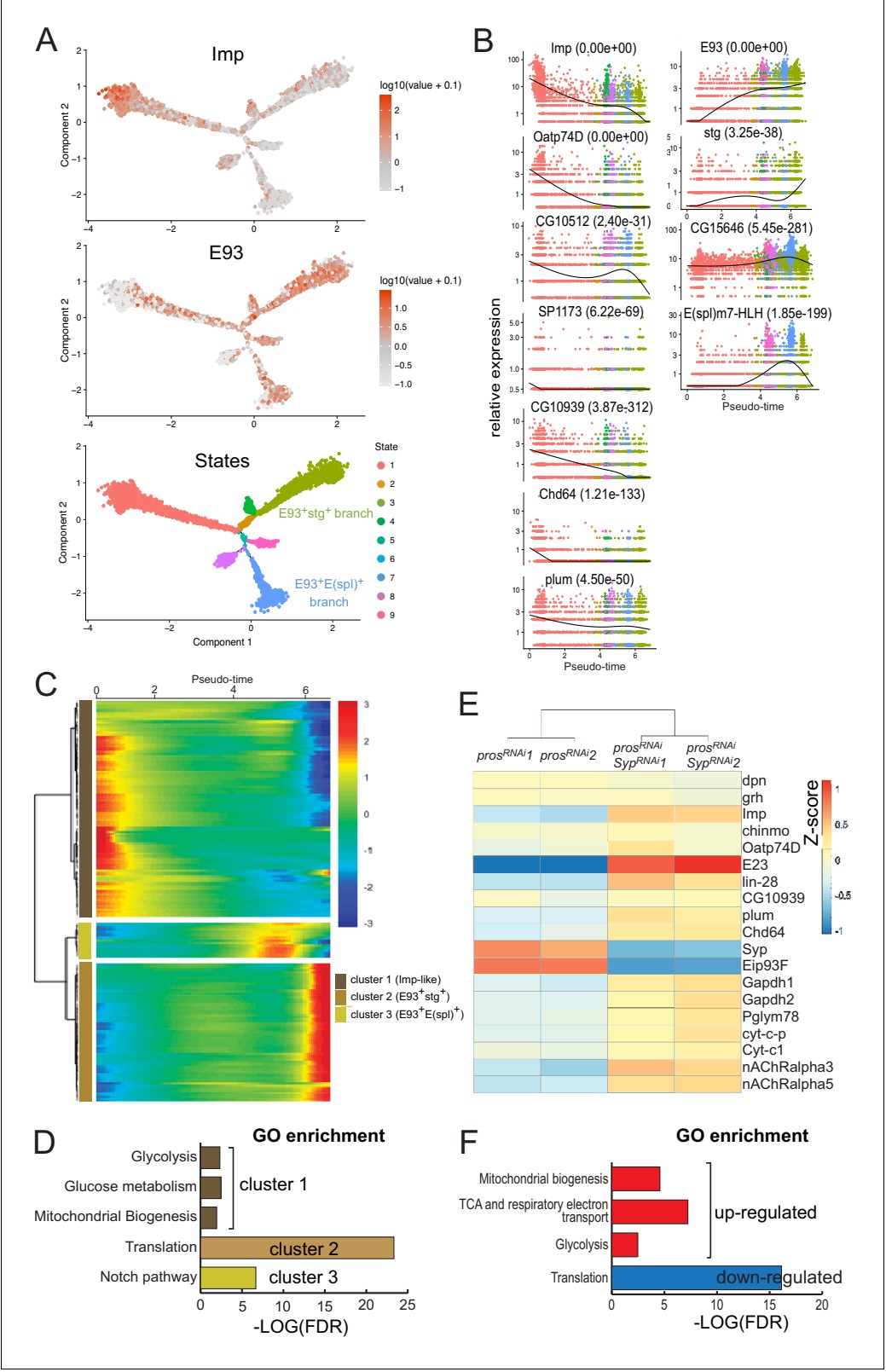

**Figure 7.** Redeployment of temporal patterning genes generates differentiation trajectories within NB tumors. (**A**) Cell trajectory reconstructed from the population of tNBs using semi-supervised pseudotime ordering. Imp+ tNBs are enriched at the root of the trajectory while E93+ tNBs are enriched in the branches terminating the trajectories. tNBs are colored according to their State along the trajectory. (**B**) Dark spline indicates levels of gene expression along the pseudotime. Cells are colored according to their State along the trajectory in (**A**). p-values for differential expression along the
*Figure 7 continued on next page*

*Figure 7 continued*

pseudotime are indicated for each gene. (C) Kinetic heatmap depicting the expression of top 200 genes that vary as a function of pseudotime. Genes are grouped in three clusters based on expression kinetics. (D) Gene ontology (GO) enrichment analysis for each cluster identified in (C). (E) Heatmap depicting enrichment of various temporal patterning, metabolic and acetylcholine receptor genes when comparing $pox^n > pros^{RNAi}$ tumors and $pox^n > pros^{RNAi}$, $Syp^{RNAi}$ tumors. (F) Gene ontology (GO) enrichment analysis when comparing the transcriptome of $pox^n > pros^{RNAi}$, $Syp^{RNAi}$ vs $pox^n > pros^{RNAi}$ tumors.

DOI: https://doi.org/10.7554/eLife.50375.019

of *stg* and *E(spl)* genes can discriminate two distinct states within the Syp⁺E93⁺ population of tNBs (*Figure 7A,B*).

To investigate whether the Imp-to-Syp transition controls the progression of transcriptional states observed along the pseudotime, we performed RNA-seq on bulk $pox^n > pros^{RNAi}$ and $pox^n > pros^{RNAi}$, $Syp^{RNAi}$ adult tumors dissected with the VNCs on which tumors grow. Between the two conditions, the expression of generic larval NB identity genes (*grh*, *dpn*) did not significantly change (*Figure 7E*). In contrast, larval NB temporal markers varied greatly, as observed with early (e.g. *Imp*, *Oatp74D*, *Sip1/CG10939*, *plum/CG6490* and *Chd64*) and late (*E93, Syp, CG15646*) markers being respectively strongly enriched and down-regulated in the $pox^n > pros^{RNAi}$, $Syp^{RNAi}$ tumors (*Figure 7E* and *Supplementary file 2*). Note that *lin-28* and *E23* were not detected in Imp⁺ tNBs with the single-cell RNA-seq approach but are detected with the bulk-seq approach, and their expression dynamics matches those of other early temporal genes such as Imp (*Figure 7E*). This transcriptomic analysis thus identifies differentiation trajectories in tumors characterized by the dynamic expression of a subset of larval temporal patterning genes that is driven by the Imp-to-Syp switch.

## The temporal trajectory governs the metabolic and proliferative properties of tNBs

In addition to temporal patterning genes, single-cell and bulk RNA-seq analyses identify other genes that are downregulated along the Imp/Syp-mediated trajectory followed by tNBs. These include for example the *nicotinic acetylcholine receptors alpha3* and *alpha5* (*nAChRalpha3*, *nAChRalpha5*) whose expression has not been reported to be temporally regulated in larval NBs (*Figure 7E* and *Figure 8—figure supplement 1*). They may therefore represent genes that are specifically activated in the tumor context and enriched in the Imp⁺Chinmo⁺ CSC-like tNBs. Glycolysis, TCA cycle and respiratory electron transport chain (OXPHOS) genes are also highly expressed in Chinmo⁺Imp⁺ tNBs and down-regulated along the Imp/Syp-dependent hierarchy (*Figures 7D,E,F* and *8A,B*), a phenomenon that has not been reported in NBs during development (except for Gapdh2 expression that is high in early MB NBs) (*Figure 1—figure supplement 2D* and *Figure 1—figure supplement 3*). Highly expressed genes in the Imp⁺Chinmo⁺ tNBs also include the *Glutamate dehydrogenase* (*Gdh*) (*Figure 8A,B*). This suggests that Chinmo⁺Imp⁺ tNBs highly rely on glutamine and glucose metabolism.

A more careful investigation of differential expression along the two main branches of the pseudotime (*Figure 8—figure supplement 2* and *Supplementary file 3*) indicates that the E93⁺stg⁺ branch is characterized by medium to low levels of glucose metabolism genes, when compared to early pre-branched cells (cluster 4), and a burst of cell cycle genes (cluster 4) followed by increased expression of ribosomal genes (cluster 3) (*Figure 8—figure supplement 2*). These data suggest that tNBs in the E93⁺stg⁺ branch exhibit transient cell cycle activity before progressing towards the end of their differentiation trajectory. In contrast, the E93⁺E(spl)⁺ branch exhibit low levels of glycolytic and cell cycle genes (cluster 4) all along (*Figure 8—figure supplement 2* and *Figure 8A*). Thus, *E(spl)* genes appear to discriminate non-proliferative tNBs that are also possibly quiescent given their low levels of metabolic gene expression. Altogether, this transcriptomic and genetic analyses suggest that the Imp-to-Syp transition in tNBs induces a cell-autonomous down-regulation of glutamine and glucose metabolism genes and the production of two different types of E93⁺ tNBs with distinct metabolic and proliferative properties, consistent with the predictions of the numerical model.

To test the importance of glucose metabolism genes during tumor growth, we knocked down *Gapdh1* and *Pglym78* (glycolysis), as well as *Cyt-c-p* and *Cyt-c1* (OXPHOS), in $pox^n > pros^{RNAi}$ tumors, all being significantly enriched in $pox^n > pros^{RNAi}$, $Syp^{RNAi}$ tumors and downregulated along

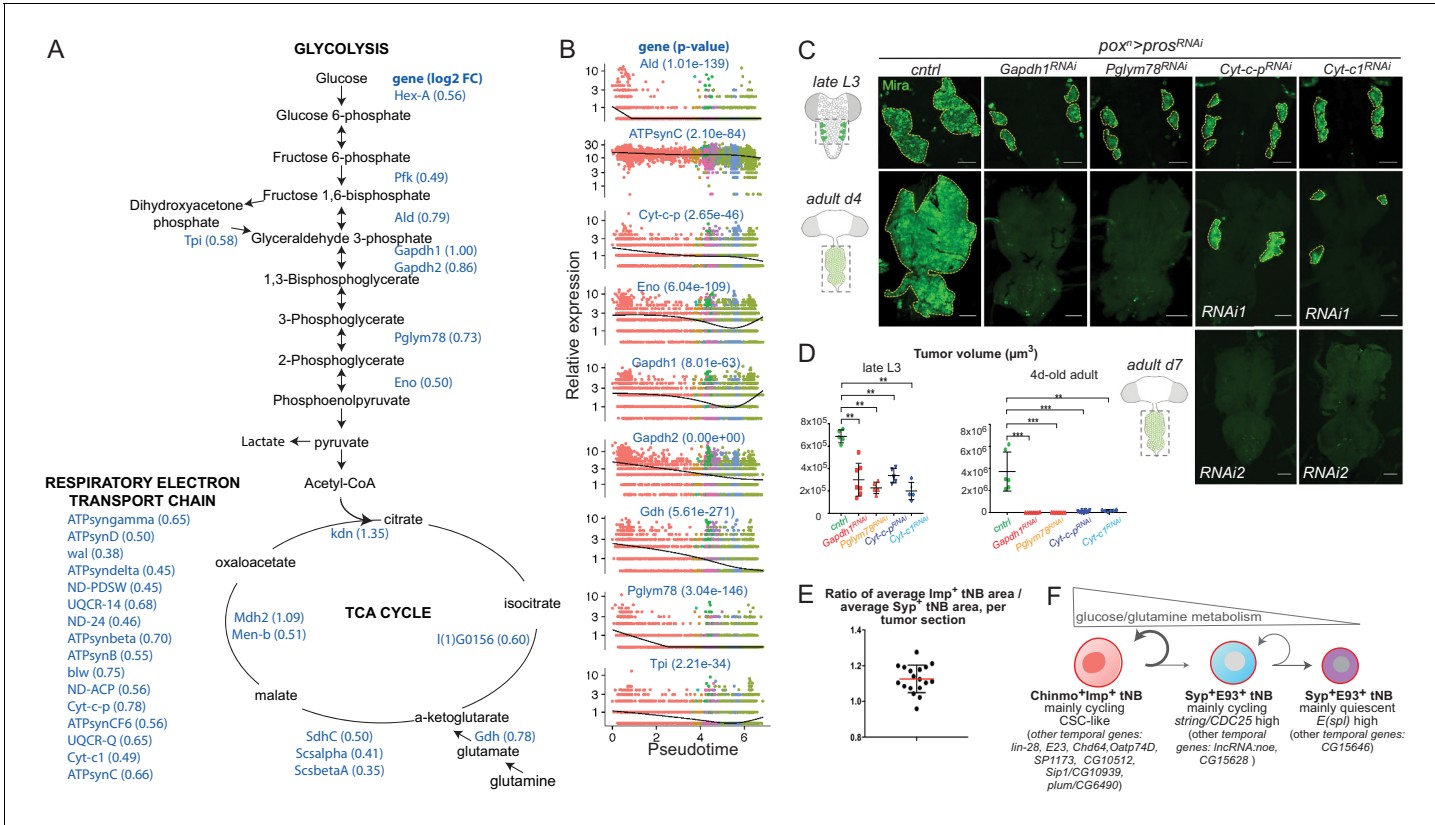

**Figure 8.** The Imp-to-Syp transition triggers down-regulation of glutamine and glucose metabolism genes along the differentiation trajectory to reduce tNB growth and proliferative potential. (**A**) Glucose metabolism pathways (glycolysis, TCA cycle and respiratory electron transport chain). Genes whose expression is enriched in $pox^n > pros^{RNAi}$, $Syp^{RNAi}$ tumors compared to $pox^n > pros^{RNAi}$ tumors are highlighted in blue. Log2 fold change is indicated in brackets. (**B**) Expression dynamics of various glycolytic, respiratory electron transport chain and glutamine metabolism genes along the tumor pseudotime. p-values for differential expression along the pseudotime are indicated for each gene. (**C**) $pox^n > pros^{RNAi}$ NB tumors are marked with anti-Mira (green). Silencing of glycolytic ($Gapdh-1^{RNAi}$, $Pglym78^{RNAi}$) or respiratory genes ($Cyt-c-p^{RNAi}$, $Cyt-c1^{RNAi}$) leads to smaller $pox^n > pros^{RNAi}$ tumors in late L3 larvae. Silencing of glycolytic or respiratory genes arrests tumor growth in 4-day-old and 7-day-old adults respectively. (**D**) Volumes of $pox^n > pros^{RNAi}$ tumors (n = 5 VNC), $pox^n > pros^{RNAi}$, $Gapdh-1^{RNAi}$ tumors (n = 7 VNC); $pox^n > pros^{RNAi}$, $Pglym78^{RNAi}$ tumors (n = 5 VNC); $pox^n > pros^{RNAi}$,$Cyt-c-p^{RNAi}$ tumors (n = 5 VNC); and $pox^n > pros^{RNAi}$,$Cyt-c1^{RNAi}$ tumors (n = 5 VNC) in late L3. $P_{Cntrl/Gapdh-1RNAi}$ = 0,002. $P_{Cntrl/Pglym78RNAi}$ = 0,007. $P_{Cntrl/Cyt-c-pRNAi}$= 0,007. $P_{Cntrl/Cyt-c1RNAi}$ = 0,007. Volumes of $pox^n > pros^{RNAi}$ tumors (n = 6 VNC); $pox^n > pros^{RNAi}$, $Gapdh-1^{RNAi}$ tumors (n = 8 VNC); $pox^n > pros^{RNAi}$, $Pglym78^{RNAi}$ tumors (n = 9 VNC); $pox^n > pros^{RNAi}$, $Cyt-c-p^{RNAi}$ tumors (n = 10 VNC); and $pox^n > pros^{RNAi}$, $Cyt-c1^{RNAi}$ tumors (n = 6 VNC) in 4-day-old adults. $P_{Cntrl/Gapdh-1RNAi}$ = 0,0007. $P_{Cntrl/Pglym78RNAi}$ = 0,0002. $P_{Cntrl/Cyt-c-pRNAi}$ = 0,0002. $P_{Cntrl/Cyt-c1RNAi}$ = 0,002. Scale bars, 50 μm. (**E**) Each dot represents the average area of Chinmo+Imp+ tNBs over the average area of Syp+E93+ tNBs for a single confocal section of a $pox^n > pros^{RNAi}$ NB tumor in a 4-day-old adult. (**F**) Schematic representation of the tumor hierarchy. Chinmo+Imp+ tNBs are proliferative and tend to self-renew. However, they can also engage in a differentiation process triggered by the Imp-to-Syp transition and involving a subset of larval temporal genes. Syp+E93+ tNBs can self-renew but also tend to exit the cell-cycle as they decrease expression of glucose/glutamine metabolism genes. Syp+E93+ tNBs, that have exited the cell-cycle, express genes of the E(spl) family. The mechanisms controlling the Imp-to-Syp transition in NB tumors, to determine the CSC proportion are unknown.

DOI: https://doi.org/10.7554/eLife.50375.020

The following figure supplements are available for figure 8:

**Figure supplement 1.** nAChRalpha3 and nAChRalpha5 are expressed in tNBs positioned at the beginning of the pseudotime.
DOI: https://doi.org/10.7554/eLife.50375.021

**Figure supplement 2.** Branch analysis of the single-cell trajectory.
DOI: https://doi.org/10.7554/eLife.50375.022

**Figure supplement 3.** Segmentation of NB tumors Segmented confocal section of a tumor using Tissue Analyzer.
DOI: https://doi.org/10.7554/eLife.50375.023

the pseudotime (p-values=8,01e$^{-63}$; 3,04e$^{-146}$; 2,65e$^{-46}$; 1,45e$^{-26}$ respectively) (**Figure 8A,B**). In all knockdown conditions, tumors underwent a slower growth rate already detectable in late L3 (**Figure 8C,D**). Moreover, tumors systematically failed to be maintained in adults showing that the expression of both glycolytic and OXPHOS genes are necessary for long-term propagation of NB tumor growth (**Figure 8C,D**). In addition, when comparing the size of tNBs within $pox^n > pros^{RNAi}$ tumors, we found that Syp$^+$E93$^+$ tNBs exhibited a smaller size than Chinmo$^+$Imp$^+$ tNBs in agreement with reduced glycolysis decreasing cell growth (**Figure 8E** and **Figure 8—figure supplement 3**). Together, this suggests that the Imp-to-Syp transition induces a metabolic switch that prevents long-term self-renewal and reduces cell growth as tNBs move down the hierarchy (**Figure 8F**). Thus, recapitulation of temporal patterning provides a tumor-intrinsic mechanism that generates metabolic heterogeneity leading totNBs with different growth properties..

## Discussion

Our study demonstrates that temporal patterning, not only determines which cells are susceptible to cancer transformation during development (**Narbonne-Reveau et al., 2016**), but also has an overarching role in governing different aspects of CNS tumor organization such as hierarchy, heterogeneity and the proliferative properties of the different types of cells via the regulation of their metabolism.

Given the recent discovery that temporal patterning is conserved in the developing mammalian brain (**Telley et al., 2019**), our study could shed light on an ancestral mechanism that governs the progression of CNS tumors with developmental origins.

### Subversion of a subset of larval NB temporal patterning genes triggers predictable hierarchical neural tumors

The rules governing the initiation and progression of CNS pediatric tumors that often exhibit stable genomes are still unclear. We had previously demonstrated that temporal patterning in *Drosophila* larval NBs delineates a window of time during which the Chinmo/Imp oncogenic module is expressed and makes early larval NBs prone to malignant transformation (**Narbonne-Reveau et al., 2016**). Here we find that after tumor initiation, temporal patterning is partly recapitulated in tNBs where it generates differentiation trajectories to constrain tumor composition and growth. This is illustrated by the presence of about 20 genes (*Imp*, *chinmo*, *Lin-28*, *E23*, *Oatp74D*, *Gapdh2*, *Sip1/CG10939*, *plum/CG6490*, *SP1173*, *Chd64*, *CG10512*, *CG44325*, *CG5953*, *Syp*, *E93*, *lncRNA:noe*, *CG15646* and *stg*), previously identified to be temporally regulated in some larval NBs, that are differentially regulated along the pseudotime/differentiation trajectory reconstructed from single-cell RNA-seq analysis of tNBs, and/or differentially expressed in Imp$^+$ vs Syp$^+$ tNBs. Thus, we identify here what appears to be a subset of a core temporal patterning program encoded in central brain and ventral nerve cord NBs that becomes deregulated upon asymmetric-division defects during early development.

Notably, the larval temporal transcription factor Cas and Svp, known to schedule the Imp-to-Syp transition during development are not enriched in Imp$^+$ tNBs suggesting that they do not play a role in regulating the Imp-to-Syp transition along the trajectory in tumors. Interestingly, while Syp is transcriptionally regulated in larval NBs, it seems rather post-transcriptionally regulated in tNBs as Syp RNAs are present throughout all clusters. This suggests that different mechanisms may be operating in tumors than during development to regulate the Imp-to-Syp transition.

We observed that the proportions of Imp$^+$ and Syp$^+$ tNBs systematically reach an equilibrium over a few days with a 20/80 (+ /- 10) ratio in $pox^n > pros^{RNAi}$ tumors. This suggests that the regulation of the Imp-to-Syp transition in tumors is not random and the predictability of the final proportions possibly implies robust underlying constraints. By investigating the population dynamics of Imp$^+$ and Syp$^+$ tNBs in $pros^{RNAi}$ tumors, we have deciphered a finely tuned hierarchical division scheme that appears to constrain the growth and cellular heterogeneity of the tumor. We showed that Imp$^+$ tNBs in the tumorigenic context favor a symmetric self-renewing mode of divisions (in more than 60% of divisions) while unlikely to exit the cell-cycle. This allows the perpetuation of a small subset of Imp$^+$ tNBs that are endowed with a seemingly unlimited self-renewing potential by the Imp/Chinmo module. Imp$^+$ tNBs can also make symmetric and asymmetric divisions that generate Syp$^+$ tNBs, leading to the production of a population of Syp$^+$E93$^+$ tNBs that accumulates

through limited self-renewal, and have a high propensity for exiting the cell-cycle. Moreover, we could show that Syp$^+$E93$^+$ tNBs are unable to generate Imp$^+$ tNBs, demonstrating a rigid cellular hierarchy reminiscent of development. In addition, in this context, Syp acts as a tumor suppressor by limiting tNB proliferation while Imp acts as an oncogene by promoting tNB proliferation and propagation of tumor growth. Together, these data argue for a scenario where cooption of the Imp-to-Syp transition is responsible for installing a hierarchical mode of tumor growth with Imp$^+$ tNBs propagating unlimited growth in a CSC-like manner, while Syp$^+$E93$^+$ tNBs acts as transient amplifying progenitors with limited self-renewing abilities. Although the Imp/Syp RNA-binding proteins have an essential and antagonistic role in governing the proliferative properties of tumor cells, the function of the other redeployed temporal patterning genes is unknown (except for chinmo, downstream to Imp and Syp, that is essential for tumor growth). As many are linked with the Imp$^+$ tNB state, it will be important in the future to decipher how they contribute to establish or maintain the CSC-like identity.

The division parameters defined by our clonal analysis and modeling approach could capture both the hierarchical aspect of tumor growth as well as the global population dynamics: from an initial homogenous pool of larval Imp+ tNBs to the stable heterogeneity observed during adulthood. It could also resolve the paradoxical observation that Chinmo+Imp+ tNBs end up in minority despite exhibiting a higher average mitotic rate. Although, like all models, we don't expect our model to perfectly recapitulate all the parameters regulating tumor growth and heterogeneity (for example, we have neglected apoptosis and neuronal differentiation that occur at low levels), we think it provides a reasonable and useful ground on which further studies can be performed for a more detailed understanding. On these lines, while the division pattern we have described with our numerical model provides estimates of division probabilities in $pox^n > pros^{RNAi}$ tumors, it says nothing as to how these probabilities are biologically set within the tumor. A possible scenario is that cell fate determination upon division relies on signals received by immediate neighboring tumor cells, resulting in effective probabilities at the scale of the whole tumor. Such a micro-environment dependent regulation of the Imp-to-Syp transition in tumors would strongly contrast with the cell-intrinsic regulation of the Imp-to-Syp transition that systematically occurs in NBs around early L3. Future studies will aim at deciphering the mechanisms that interfere with the developmental progression of the temporal patterning, upon asymmetric-division defects, to favor the self-renewing mode of divisions undergone by the Chinmo$^+$Imp$^+$ tNBs, allowing perpetuation of a population of CSC-like cells.

Noteworthy, $pros^{RNAi}$ and $snr1/dSmarcb1^{RNAi}$ tumors exhibit different but reproducible ratios of Imp$^+$ and Syp$^+$ tNBs. This suggests the existence of tumor-specific mechanisms that fine-tune the Imp-to-Syp transition. Such mechanisms may be related to the tumor cell of origin, or to the genetic insult that initiated NB amplification. Further analysis will help identifying tumor-intrinsic signals regulating the balance between Chinmo$^+$Imp$^+$ tNBs and Syp$^+$E93$^+$ tNBs in various types of NB tumors.

Until recently, the existence of temporal patterning in mammalian neural progenitors remained uncertain. Elegant single-cell transcriptomic studies of embryonic cortical and retinal progenitors in mice have now revealed that they transit through different transcriptional states that are transmitted to their progeny to generate neuronal diversity, similar to temporal patterning in *Drosophila* (*Clark et al., 2019*; *Telley et al., 2019*). However, it remains unknown whether temporal patterning determines the cell of origin and governs the growth of CNS tumors in children. Along these lines, the finding that the transcriptional programs operating in cerebellar progenitors during fetal development are recapitulated in medulloblastomas is promising (*Vladoiu et al., 2019*). By uncovering the overarching role of temporal patterning in governing tumor susceptibility during CNS development and in constraining tumor properties during cancer progression in *Drosophila*, our work thus possibly provides a new conceptual framework to better understand CNS tumors in children.

## Tumor-intrinsic regulation of NB metabolism by the temporal patterning system

Because of the difficulty to investigate metabolism at the single-cell level, it has been difficult to determine how heterogeneous is the metabolic activity of cells in tumors, and how it is controlled. Here, using a combination of single-cell and bulk RNA-seq approaches, we find that progression of temporal patterning provides a tumor-intrinsic mechanism that generates heterogeneity in the proliferative abilities of tumor cells through the progressive silencing of glucose and glutamine metabolism genes.

Consequently, Chinmo⁺Imp⁺ tNBs, that lie at the top of the hierarchy, highly express glycolytic and respiratory/OXPHOS genes, as well as Gdh, that are down-regulated by the Imp-to-Syp transition. This default high expression of both glutamine and glucose metabolism genes in CSC-like Chinmo⁺Imp⁺ tNBs likely favors sustained self-renewal, but could also confer plasticity and a way to adapt cellular metabolism to different environmental conditions as frequently observed in CSCs (e.g. glutamine can compensate for glucose shortage) (*Sancho et al., 2016*).

We showed that Syp⁺E93⁺ tNBs exhibit a reduced size, and that knock-down of glycolytic (*Gapdh1* or *Pglym78)* or respiratory/OXPHOS genes (*Cyt-c-p* or *Cyt-C1*) prevented propagation of tumor growth in adults. Thus, reduction of biosynthesis and energy production through down-regulation of glucose and glutamine metabolism genes after the Imp-to-Syp transition could progressively exhaust Syp⁺E93⁺ tNB growth and self-renewing ability, ultimately leading to cell-cycle exit.

With our demonstration that temporal patterning regulates glycolytic, TCA cycle and OXPHOS genes in NB tumors, our work provides a tumor-intrinsic mechanism that creates metabolic heterogeneity to control the proliferative potential of the various tumor cells. We have observed that Syp⁺E93⁺ tNBs associated with lowest levels of metabolic and cell-cycle genes also upregulate genes of the *E(spl)* genes. Interestingly, expression of *Hes* genes (orthologs of *Enhancer of split* genes) in vertebrate neural stem cells is associated with the maintenance of a quiescent state in adults (*Chapouton et al., 2011*; *Sueda et al., 2019*). Thus, E(spl) genes may promote the quiescent tNB state identified with our clonal and numerical analysis while preventing their differentiation in neurons.

Down-regulation of the mRNA levels of metabolic genes after the Imp-to-Syp transition could be due to the silencing of a transcriptional activator or to an increased mRNA degradation. On one hand, Chinmo is a likely candidate for the first scenario, as its inactivation reduces growth in NBs (*Narbonne-Reveau et al., 2016*) and we showed that it is a direct target of both Imp and Syp. On the other hand, the second scenario is consistent with Imp orthologs in human being able to promote OXPHOS and proliferation in glioma cells, through the post-transcriptional regulation of mitochondrial respiratory chain complex subunits (*Janiszewska et al., 2012*).

We have also identified a small population of tNBs expressing various stress or growth arrest factors. One of these factors, Xrp1, is a transcriptional target of *p53* in the response to irradiation. *Xrp1* expression has also recently been linked to defects in translation rates, together with the expression of *Irbp18* and *GstE6* (*Lee et al., 2018*). Thus, these factors may label a subset of tNBs undergoing DNA or translational stress. The reason and consequences of such cellular stresses in tumor progression need to be further investigated.

Our transcriptomic analyses have revealed strong similarities in the differentiation trajectories of tNBs in tumors and of NBs in larvae. Yet, it is surprising that the down-regulation of glutamine and glucose metabolism genes has not been detected in NBs during larval development, after the Imp-to-Syp transition (*Ren et al., 2017*). It is possible that the glial niche surrounding NBs, that is known to influence NB growth properties during larval stages (*Chell and Brand, 2010*; *Cheng et al., 2011*; *Sousa-Nunes et al., 2011*), somehow sustains high levels of glucose metabolism genes in late Syp⁺E93⁺ NBs. Given that this glial niche is absent in tumors, Syp⁺E93⁺ tNBs may not be able to sustain the high expression of metabolic genes imposed by the Imp/Chinmo module, leading to progressive cell-cycle exit.

## Imp and syp: a conserved couple of antagonistic RNA-binding proteins regulating cell hierarchy in human tumors?

Chinmo and Imp are reminiscent to oncofetal genes in mammals, in that their expression decrease rapidly as development progresses while they are mis-expressed in tumors. Along these lines, the three IMP orthologs in humans (also called IGF2BP1-3) are also known as oncofetal genes. They emerge as important regulators of cell proliferation and metabolism in many types of cancers including pediatric neural cancers (*Bell et al., 2015*; *Dai et al., 2017*; *Degrauwe et al., 2016a*; *Degrauwe et al., 2016b*; *Janiszewska et al., 2012*). Along evolution, the ancestral *Syncrip* gene has been subjected to several rounds of duplication and has diverged into five paralogs in mammals, some of them emerging as tumor suppressors with an important role in tumor progression (*Sakurai et al., 2016*; *Vanharanta et al., 2014*).

Thus, the respective oncogenic and tumor suppressor roles of IMP and SYNCRIP gene families appear to have been conserved in humans and they may not be restricted to tumors of neural origin.

Our study therefore raises the exciting possibility that these two families of RNA-binding proteins form a master module at the top of the self-renewal/differentiation cascades, that regulates CSC populations and hierarchy in a spectrum of human cancers.

# Materials and methods

## Key resources table

| Reagent type (species) or resource | Designation | Source or reference | Identifiers | Additional information |
|---|---|---|---|---|
| Genetic reagent (*D. melanogaster*) | *UAS-Flybow. 1.1* | Bloomington Drosophila Stock Center (*Hadjieconomou et al., 2011*) | RRID:BDSC_35537 | |
| Genetic reagent (*D. melanogaster*) | *UAS-mCherry*<sup>chinmoUTRs</sup> | Cédric Maurange (*Dillard et al., 2018*) | | *mCherry* reflects the post-transcriptional regulation of *chinmo*. |
| Genetic reagent (*D. melanogaster*) | *UAS-Syp*<sup>RNAi1</sup> | VDRC | 33011 | RNAi1 |
| Genetic reagent (*D. melanogaster*) | *UAS-Syp*<sup>RNAi2</sup> | VDRC | 33012 | RNAi2 |
| Genetic reagent (*D. melanogaster*) | *UAS-Syp-RB-HA* | Tzumin Lee (*Liu et al., 2015*) | | |
| Genetic reagent (*D. melanogaster*) | *UAS-pros*<sup>RNAi1</sup> | Bloomington Drosophila Stock Center | RRID:BDSC_26745 | RNAi1 |
| Genetic reagent (*D. melanogaster*) | *UAS-pros*<sup>RNAi2</sup> | VDRC | 101477 | RNAi2 |
| Genetic reagent (*D. melanogaster*) | *UAS-Imp*<sup>RNAi</sup> | VDRC | 20322 | |
| Genetic reagent (*D. melanogaster*) | *UAS-chinmo*<sup>RNAi</sup> | Bloomington Drosophila Stock Center | RRID:BDSC_33638 | |
| Genetic reagent (*D. melanogaster*) | *UAS-dicer2* | Bloomington Drosophila Stock Center | RRID:BDSC_24650 RRID:BDSC_24651 | was used in combination with GAL4 lines in order to improve RNAi efficiency. |
| Genetic reagent (*D. melanogaster*) | *UAS-mCD8::GFP* | Bloomington Drosophila Stock Center | RRID:BDSC_5130 RRID:BDSC_32185 | |
| Genetic reagent (*D. melanogaster*) | *UAS-myr::GFP* | Bloomington Drosophila Stock Center | RRID:BDSC_32197 | |
| Genetic reagent (*D. melanogaster*) | *UAS-mCherry.NLS* | Bloomington Drosophila Stock Center | RRID:BDSC_38424 | |
| Genetic reagent (*D. melanogaster*) | *UAS-Snr1*<sup>RNAi1</sup> | VDRC | 108599 | RNAi1 |
| Genetic reagent (*D. melanogaster*) | *UAS-Snr1*<sup>RNAi2</sup> | VDRC | 32372 | RNAi2 |
| Genetic reagent (*D. melanogaster*) | *UAS-Gapdh1*<sup>RNAi</sup> | VDRC | 100596 | |
| Genetic reagent (*D. melanogaster*) | *UAS-Pglym78* | VDRC | 106818 | |
| Genetic reagent (*D. melanogaster*) | *UAS-Cyt-c1*<sup>RNAi1</sup> | VDRC | 9180 | RNAi1 |

*Continued on next page*

*Continued*

| Reagent type (species) or resource | Designation | Source or reference | Identifiers | Additional information |
|---|---|---|---|---|
| Genetic reagent (*D. melanogaster*) | UAS-Cyt-c1RNAi2 | VDRC | 109809 | RNAi2 |
| Genetic reagent (*D. melanogaster*) | UAS-Cyt-c-pRNAi1 | VDRC | 33019 | RNAi1 |
| Genetic reagent (*D. melanogaster*) | UAS-Cyt-c-pRNAi2 | VDRC | 106759 | RNAi2 |
| Genetic reagent (*D. melanogaster*) | w, tub-GAL4, UAS-nlsGFP::6xmyc::NLS, hsFLP122; FRT82B, tubP-GAL80/TM6B | Cédric Maurange (*Narbonne-Reveau et al., 2016*) | | MARCM line |
| Genetic reagent (*D. melanogaster*) | FRT82B snr16C hdac36C | Bloomington Drosophila Stock Center (*Koe et al., 2014*) | RRID:BDSC_34494 | |
| Genetic reagent (*D. melanogaster*) | Imp-GFP | Florence Besse | | protein trap line |
| Genetic reagent (*D. melanogaster*) | poxn-Gal4 | Bloomington Drosophila Stock Center (*Boll and Noll, 2002*) | RRID:BDSC_ 66685 | |
| Genetic reagent (*D. melanogaster*) | ase-Gal80, wor-GAL4, UAS-dcr2 | Juergen Knoblich (*Eroglu et al., 2014*) | | |
| Genetic reagent (*D. melanogaster*) | nab-GAL4 | Kyoto DGRC | 6190 | |
| Genetic reagent (*D. melanogaster*) | UAS-FLP, Ubi-p63EFRTstopFRTnEGFP | Bloomington Drosophila Stock Center (*Evans et al., 2009*) | RRID:BDSC_28282 | G-trace |
| Genetic reagent (*D. melanogaster*) | hs-mFLP5 | Bloomington Drosophila Stock Center (*Hadjieconomou et al., 2011*) | RRID:BDSC_35534 | Flipase for Flybow |
| Antibody | chicken polyclonal anti-GFP | Aves Labs #GFP-1020 | RRID:AB_10000240 | 1:1000 |
| Antibody | rabbit polyclonal anti-RFP | Rockland #600-401-379 | RRID:AB_2209751 | 1:500 |
| Antibody | rat monoclonal anti-RFP | Chromotek #5F8 | RRID:AB_2336064 | 1:500 |
| Antibody | mouse monoclonal anti-Miranda | Alex Gould | | 1:50 |
| Antibody | rabbit polyclonal anti-PH3 | Millipore #06–570 | RRID:AB_310177 | 1:500 |
| Antibody | rat monoclonal anti-PH3 | Abcam #AB10543 | RRID:AB_2295065 | 1:500 |
| Antibody | rat monoclonal anti-Elav | DSHB | #9F8A9 | 1:50 |
| Antibody | rabbit polyclonal anti-cleaved Dcp-1 | Cell Signaling #9578 | RRID:AB_2721060 | 1:500 |
| Antibody | rat polyclonal anti-Chinmo | Nick Sokol | | 1:500 |
| Antibody | rabbit polyclonal anti-Imp | Paul Macdonald | | 1:500 |
| Antibody | guinea pig polyclonal anti-Syp | Ilan Davis | | 1:500 |

*Continued*

| Reagent type (species) or resource | Designation | Source or reference | Identifiers | Additional information |
|---|---|---|---|---|
| Antibody | Rabbit polyclonal anti-Syp | Ilan Davis | | 1/200 |
| Antibody | rabbit polyclonal anti-E93 | Daniel J. McKay | | 1/2500 |
| Antibody | guinea-pig polyclonal antibody | Bill McGinnis | | 1/200 |
| Chemical compound, drug | Dapi | Vector Laboratories Cat# H-1400 | RRID:AB_2336787 | |
| Recombinant DNA reagent | Chinmo cDNA clone | DGRC, EST Collection | #RE59755 | |
| Sequence-based reagent | Chin-5'UTR/pBS_Forward | This paper | PCR primers | Described in Affinity pull-down assays section in the Materials and methods |
| Sequenced-based reagent | Chin-5'UTR/pBS_Reverse | This paper | PCR primers | Described in Affinity pull-down assays section in the Materials and methods |
| Sequenced-based reagent | Chin-3'UTR/pBS_Forward | This paper | PCR primers | Described in Affinity pull-down assays section in the Materials and methods |
| Sequenced-based reagent | Chin-3'UTR/pBS_Reverse | This paper | PCR primers | Described in Affinity pull-down assays section in the Materials and methods |
| Software, algorithm | Seurat and Monocle codes used for single-cell RNA-seq data analysis | This paper | https://github.com/cedricmaurange/Genovese-et-al.-2019 | See the Single-cell mRNA sequencing and analysis section in the Materials and methods |
| Software, algorithm | Code used for numerical model | This paper | http://dx.doi.org/10.17632/j2j9gmyb6m.1 | See the Numerical model section in the Materials and methods |

## Fly culture

*Drosophila* stocks were maintained at 18°C on standard medium (8% cornmeal/8% yeast/1% agar).

## Fly lines

The protein trap line used was:

- Imp-GFP (Bloomington Stock Centre #G0080)

The Gal4 lines used were:

- *pox^n-Gal4* is active in six homologous thoracic NBs of the VNC (***Boll and Noll, 2002***).
- *ase-Gal80, wor-GAL4, UAS-dcr2* has been crossed with various *UAS-RNAi* transgenes for specific mis-expression in type II NBs of the central brain (***Eroglu et al., 2014***).
- *nab-Gal4, UAS-dicer2, UAS-GFP. nab-GAL4* (#6190 from Kyoto DGRC) is a GAL4 trap inserted into *nab* (CG33545) that is active in all NBs of the VNC and central brain from late embryogenesis

The UAS lines used were:

- *UAS-Flybow. 1.1* (Bloomington Stock Centre #35537) (*Hadjieconomou et al., 2011*).
- *UAS-mCherry$^{chinmoUTRs}$* (*Dillard et al., 2018*). In NB tumors carrying *UAS-mCherry$^{chinmoUTRs}$*, expression of *mCherry* reflects the post-transcriptional regulation of endogenous *chinmo*. Consistently, we observe that mCherry always overlaps with endogenous Chinmo as detected by immunostaining (*Figure 1—figure supplement 4*).
- *UAS-Syp$^{RNAi1}$*(Vienna Drosophila RNAi Center #33011)
- *UAS-Syp$^{RNAi2}$*(Vienna Drosophila RNAi Center #33012)
- *UAS-Syp-RB-HA* (T Lee, Janelia Research Campus, Virginia, USA) (*Liu et al., 2015*).
- *UAS-pros$^{RNAi1}$* (Transgenic RNAi Project #JF02308, Bloomington Stock Centre #26745)
- *UAS-pros$^{RNAi2}$* (Vienna Drosophila RNAi Center #101477)
- *UAS-Imp$^{RNAi}$* (Vienna Drosophila RNAi Center #20322)
- *UAS-chinmo$^{RNAi}$* (Transgenic RNAi Project #HMS00036, Bloomington Stock Centre #33638)
- *UAS-dicer2* (Bloomington Stock Centre #24650 and #24651) was used in combination with GAL4 lines in order to improve RNAi efficiency.
- *UAS-mCD8::GFP* (Bloomington Stock Centre #5130 and #32185) and *UAS-myr::GFP* (Bloomington Stock Centre *#32197)* were used to follow the driver expression.
- *UAS-mCherry.NLS* (Bloomington Stock Centre #38424)
- *UAS-Imp$^{RNAi}$; UAS-Syp$^{RNAi2}$ (UAS-Syp$^{RNAi2}$* from Vienna Drosophila RNAi Center #33012)
- *UAS-Syp$^{RNAi1}$; UAS-chinmo$^{RNAi}$ (UAS-Syp$^{RNAi1}$* from Vienna Drosophila RNAi Center #33011)
- *UAS-pros$^{RNAi1}$, UAS-mCD8::GFP (UAS-pros$^{RNAi1}$* from Transgenic RNAi Project #JF02308)
- *UAS-pros$^{RNAi2}$; UAS-mCherry$^{chinmoUTRs}$(UAS-pros$^{RNAi2}$* Vienna Drosophila RNAi Center #101477)
- *UAS-Snr1$^{RNAi-1}$* (Vienna Drosophila RNAi Center #108599).
- *UAS-Snr1$^{RNAi-2}$* (Bloomington Stock Centre #32372).
- *UAS-Gapdh1$^{RNAi}$* (Vienna Drosophila RNAi Center #100596)
- *UAS-Pglym78 $^{RNAi}$* (Vienna Drosophila RNAi Center #106818)
- *UAS-Cyt-c1$^{RNAi1}$* (Vienna Drosophila RNAi Center #9180)
- *UAS-Cyt-c1$^{RNAi2}$* (Vienna Drosophila RNAi Center #109809)
- *UAS-Cyt-c-p$^{RNAi1}$* (Vienna Drosophila RNAi Center #33019)
- *UAS-Cyt-c-p$^{RNAi2}$* (Vienna Drosophila RNAi Center #106759)

The MARCM and FRT stocks for used were:

- *w, tub-GAL4, UAS-nlsGFP::6xmyc::NLS, hsFLP$^{122}$; FRT82B, tubP-GAL80/TM6B* (*Narbonne-Reveau et al., 2016*)
- *FRT82B snr1$^{6C}$ hdac3$^{6C}$* (*Koe et al., 2014*)

Fly crosses were set up and raised at 29°C. To label all tumor cells with GFP, *pox$^n$-Gal4, UAS-pros$^{RNAi}$, UAS-dicer2* flies were crossed with the G-TRACE system *UAS-FLP, Ubi-p63E$_{FRT}$stop$_{FRT}$-nEGFP* (Bloomington Stock Centre #28282) (*Evans et al., 2009*). To generate mCherry$^+$ clones in tumors, *pox$^n$-GAL4, UAS-pros$^{RNAi1}$, UAS-mCD8::GFP; UAS-dicer2* flies were crossed to *UAS-Flybow.1.1; hs-mFLP5* (Bloomington Stock Centre #35534 and #35537) (*Hadjieconomou et al., 2011*). The progeny of this cross was collected 48 hours after adult eclosion and heat-shocked for 1h30' at 37°C. Flies were put back at 29°C after heat-shock. Note that differentiated neurons generated from mCherry$^+$ tNBs will not retain mCherry expression due to the silencing of *pox$^n$-GAL4* in neurons. MARCM clones were induced by heat-shocking 1 hr at 37°C larvae that have just hatched.

## Image processing

Confocal images were acquired on a Zeiss LSM780 microscope. FIJI-ImageJ was used to process confocal data and to compile area and volume data. Imaris Image Analysis Software (http://bitplane.com) was used to generate 3D representations of clones in tumors.

## Tumor segmentation using tissue analyzer

For *Figure 8E*, *pox$^n$ > pros$^{RNAi}$* tumors carrying an Imp-GFP marker were dissected, fixed in 4% paraformaldehyde and stained with antibodies against Miranda and GFP proteins. VNCs were visualized in a Zeiss LSM780 confocal microscope and co-planar images were made of several sections ranging from the tumor surface to its interior. Several confocal planes for four tumors (n = 21) were analyzed using the 'Tissue Analyzer' plugin for FIJI-ImageJ (*Aigouy et al., 2016*). This tool allows for cell membrane tracing, via the signal from Miranda which marks the cellular border of all tumor cells, segmentation and quantification of several parameters, including size, of each tumor cell individually.

Chinmo[+]Imp[+] tNBs are identified by their strong Imp-GFP signal. Initial automatic segmentation was further refined manually to ensure all cell boundaries were properly tracked and cells assigned a correct identity (*Figure 8—figure supplement 3*). Data points represent different planar sections within the same tumor, in order to account for variability within each tumor.

## Statistical analysis

For each experiment, at least three biological replicates were performed. Biological replicates are defined as replicates of the same experiment with flies being generated by different parent flies. For all experiments, we performed a Mann-Whitney test for statistical analysis. Statistical analysis was performed using GraphPad Prism version 7.04 for Windows (GraphPad Software, La Jolla California USA, www.graphpad.com). Results are presented as dot plots. Error bars represent s.d. from the mean. The sample size (n) and the *P*-value are reported in the figure legends (****$p \leq 0.0001$; ***$p \leq 0.001$; **$p \leq 0.01$; *$p \leq 0.05$; ns, not significant).

## Immunohistochemistry

Larval and adult VNCs were dissected in phosphate-buffered saline (PBS) and fixed for 10 min at room temperature (RT) in 4% paraformaldehyde/PBS. VNCs were rinsed in PBT (PBS containing 0.5% Triton X-100) and incubated with primary antibody overnight at 4°C. Secondary antibodies (Jackson ImmunoResearch) were incubated overnight at 4°C. VNCs were mounted in Vectashield (Clinisciences, France) with or without DAPI for image acquisition. The following primary antibodies were used: chicken anti-GFP (1:1000, Aves #GFP-1020), rabbit anti-RFP to label mCherry (1:500, Rockland #600-401-379), rat anti-RFP (1:500, Chromotek #5F8), mouse anti-Miranda (1:50, A Gould, Francis Crick Institute, London, UK), rabbit anti-PH3 (1:500, Millipore #06–570), rat anti-PH3 (1:500, Abcam #AB10543), rat anti-Elav (1:50, DSHB #9F8A9), rabbit anti-cleaved Dcp-1 (1:500, Cell Signaling #9578)), rat anti-Chinmo (1:500, N Sokol, Indiana University, Bloomington, USA), rabbit anti-Imp (1:500, P Macdonald), guinea pig anti-Syp (1/500, I Davis, Oxford University, UK), rabbit anti-Syp (1/200, I Davis, Oxford University, UK), rabbit anti-E93 (1/2500, DJ McKay, University of North Carolina, Chapel Hill).

## Affinity pull-down assays

Affinity pull-down assays were performed as previously described (*Medioni et al., 2014*), and Western Blots performed using rat anti-Imp and rabbit anti-Syp antibodies. The 5' and 3' UTRs of chinmo were cloned from the cDNA clone #RE59755 (DGRC, EST Collection) using the following primers:

| | |
|---|---|
| Chin-5'UTR/pBS_Forward | **aatttctaga**AGTCAAAAAGAAACTGCCGTG |
| Chin-5'UTR/pBS_Reverse | gatg**aagctt**GGTGCCAGCAGTGATGCT |
| Chin-3'UTR/pBS_Forward | taaa**tctaga**GAAGCAGCCGCAACAGCA |
| Chin-3'UTR/pBS_Reverse | tttt**aagctt**GGTGAATTTTCATTTGTACGAAGAA |

Capitals represents analogous or complementary parts of the chinmo sequence. In bold are the restriction sites used for cloned into pBS-KS(-) (TCTAGA = *Xba*I, AAGCTT = *Hin*dIII), and in lower case are extra bases for efficient digestion. This plasmid was used to generate the UTP-biotinylated-RNA probes (5'UTR and 3'UTR).

## RNA extraction of bulk tumors

To generate NB tumors *pox^n^-Gal4, UAS-mCherry.NLS; UAS-dicer2 flies* were crossed to flies carrying:

1. *UAS-pros^RNAi1^*
2. *UAS-Syp^RNAi1^;UAS-pros^RNAi1^*

Crosses were grown at 29°C. Sixteen adult females and 13 adult males (8 day-old) were killed in 70% ethanol and washed once in PBS. VNCs containing tumors were dissected in PBS and collected in a RNase-free Protein LoBinding tube filled with 750 µL of Lysis Buffer (Buffer RLT, RNeasy Mini Kit, Qiagen) supplemented with 7.5 µL of β-mercaptoethanol and 500 µL of glass beads (diameter

0.75–1 mm, Roth, A554.1). Samples were disrupted and homogenized using the tissue homogenizer Precellys 24 (Bertin Technologies).

Sample tubes were then stored at −80°C up to RNA extraction. Total RNA was extracted using the RNeasy Mini Kit (Qiagen). RNA quality and quantity were checked by running samples on an Agilent RNA 6000 Pico Chip (Agilent Technologies). For each condition, samples were made from 29 dissected VNCs pooled from adult flies born from several crosses (more than 3). For technical replicates, two samples of each condition were generated and treated in parallel until sequencing.

## Library preparation and sequencing of bulk tumors

Total RNA-Seq libraries were generated from 130 ng of total RNA using TruSeq Stranded Total RNA LT Sample Prep Kit with Ribo-Zero Gold (Illumina, San Diego, CA), according to manufacturer's instructions. Briefly, cytoplasmic and mitochondrial ribosomal RNA (rRNA) was removed using biotinylated, target-specific oligos combined with Ribo-Zero rRNA removal beads. Following purification, the depleted RNA was fragmented into small pieces using divalent cations at 94°C for 2 min. Cleaved RNA fragments were then copied into first strand cDNA using reverse transcriptase and random primers followed by second strand cDNA synthesis using DNA Polymerase I and RNase H. Strand specificity was achieved by replacing dTTP with dUTP during second strand synthesis. The double stranded cDNA fragments were blunted using T4 DNA polymerase, Klenow DNA polymerase and T4 PNK. A single 'A' nucleotide was added to the 3' ends of the blunt DNA fragments using a Klenow fragment (3' to 5'exo minus) enzyme. The cDNA fragments were ligated to double stranded adapters using T4 DNA Ligase. The ligated products were enriched by PCR amplification (30 s at 98°C; [10 s at 98°C, 30 s at 60°C, 30 s at 72°C] x 12 cycles; 5 min at 72°C). Surplus PCR primers were further removed by purification using AMPure XP beads (Beckman-Coulter, Villepinte, France) and the final cDNA libraries were checked for quality and quantified using capillary electrophoresis.

The libraries were loaded in the flowcell at a concentration of 2.8 nM and clusters were generated using the Cbot and sequenced on an Illumina HiSeq 4000 system as paired-end 100 base reads following Illumina's instructions.

## RNA-Seq analysis of bulk tumors

Quality control of raw reads was done using FastQC. Reads were mapped to *dmel6* reference genome with *Subread aligner* using default parameters (*Liao et al., 2013*). Read counts were calculated using *Subread featureCount* (*Liao et al., 2014*). Differential expression was computed using R package DESeq2 (*Love et al., 2014*). To concentrate on genes that are expressed in tNBs and remove those that are specific to wild type neurons and glia of the VNC, we excluded from the bulk RNA-seq analysis all genes that are not expressed in at least 0,5% of the 5740 cells used for the single-cell RNA-seq experiment. Among the remaining 6660 genes, we selected as significantly enriched genes those with an adjusted p-value<0001. Gene Ontology and Reactome Pathway analysis was made using *Panther* (http://www.pantherdb.org/). Sequencing data have been deposited in GEO under accession code GSE114562.

## Preparation of tNBs for single-cell mRNA-seq

NB tumors were generated in *pox^n^-Gal4, UAS-GFP UAS-pros^RNAi2^, UAS-dicer2* flies. Fifty-six adult females (6–8 day-old) were killed in 70% ethanol and washed once in PBS. VNCs were dissected in PBS, collected in a RNase-free Protein LoBinding tube filled with ice-cold PBS, and incubated in a freshly prepared dissociation solution containing 0,4% bovine serum albumin (BSA), 1 mg/mL collagenase I and papain (Sigma Aldrich) in PBS for 75 min at 29°C with lowest agitation. Tissues were then disrupted manually by pipetting up and down with a 200 µL tip. Dissociated cells were pelleted for 20 min at 300 g at 4°C to remove the dissociation solution and to resuspend cells in ice-cold PBS + 0,4% BSA. The cell suspension was filtered through a 30 µm mesh Pre-Separation Filter (Miltenyi) to remove debris and transferred in new RNase-free Protein LoBinding tube for FACS sorting. Forty thousand GFP+ tNBs cells were isolated using a FACSAriaII machine (BD) with an 85 µm nozzle, at 45 psi low pressure and according to viability, cell size and GFP intensity. In the next 30 min, sorted-cells were encapsidated using with the Single Cell Controller (10X genomics) for single-cell RNA-seq.

## Single-cell mRNA sequencing and analysis

Codes used for single-cell RNA-seq analysis are available at: https://github.com/cedricmaurange/Genovese-et-al.-2019 (*Maurange, 2019*; copy archived at https://github.com/elifesciences-publications/Genovese-et-al.-2019).

### Single-cell mRNA sequencing

Single cells were processed using the Single cell 3' Library, Gel beads and multiplex kit (10X Genomics, Pleasanton) as per the manufacer's protocol. Cells were partitioned into nanoliter-scale Gel Bead-In-Emulsions (GEMs) with the Chromium Single Cell Controller (10X genomics, Pleasanton), where all generated cDNA share a common 10x Barcode. Libraries were generated and sequenced from the cDNA and the10x Barcodes are used to associate individual reads back to the individual partitions. Analysis using molecular indexing information provides an absolute digital measurement of gene expression levels. Sequencing was performed using a NextSeq 500 Illumina device (1sample) containing transcript lengh of 57 bp. Sequencing data have been deposited in GEO under accession code GSE114986.

### Data processing

Single-cell mRNA seq data were analyzed using the 10x Genomics suite *Cell Ranger 2.0.1* with default settings for de-multiplexing, aligning reads to the *Drosophila* genome (10X Genomics pre-built *dm6* reference genome) with STAR and counting unique molecular identifiers (UMIs) to build transcriptomic profiles of individual cells. This first level of analysis generates quality metrics (Q30, number of reads by sample...), FASTQ files and filtered genes matrices.

### PC analysis, UMAP clustering and differential expression

Filtered gene matrices generated via *Cell Ranger 2.0.1* was then processed with the R package *Seurat v3.0.2*, using the online tutorial as a guide (https://satijalab.org/seurat/v3/pbmc3k_tutorial.html). We kept all cells that had between 200 and 4000 genes/cell. Cells with more than 5% mitochondrial gene expression were removed. Cell cycle genes were regressed out using the *CellCycleScoring* and *ScaleData* functions.

PCA was then utilized to determine sources of variability. Twenty significant PCs were determined based on the *JackStraw* function. We used the *PCHeatmap* function to explore gene sets representing the main sources of heterogeneity in the dataset. Cells were then clustered with the *FindClusters* function a resolution parameter of 0.5. Differentially expressed genes were determined with the *FindAllMarkers* and function. Code available upon request.

### Cell trajectory and pseudo time analysis by Single-cell mRNA sequencing

Filtered gene matrices generated via *Cell Ranger 2.0.1* was processed with the *R package Monocle* version v2.10.1. for pseudotime analysis (*Qiu et al., 2017a*; *Qiu et al., 2017b*; *Trapnell et al., 2014*). Graphics were generated through *ggplot2* R library. Analysis was performed following online tutorial as a guide (http://cole-trapnell-lab.github.io/monocle-release/docs/#constructing-single-cell-trajectories). UMI Counts were both normalized across cells using *estimateSizeFactors* function and gene expression variance was estimated using *estimateDispersions* functions to remove outlier genes using DESeq2 packages (*Love et al., 2014*). Semi-supervised ordering was performed with the *classifyCells* function. Only genes expressed in at least 10 cells were taken in account. Marker genes were selected for subsequent analysis based on their normalized average expression and variability across cells using *markerDiffTable* function. The top 200 genes were used for ordering.

Dimension reduction was performed with *Monocle* implementation of Discriminative Dimension Reduction Tree (DDRTree) algorithm (*Qi Mao et al., 2017*) using *reduceDimension* function. Minimum spanning tree computation, cell state assignations and pseudo time reconstruction was performed using *orderCells* functions. *differentialGeneTest* and *plot_pseudotime_heatmap* functions were used to uncover the top 200 genes that change as a function of pseudotime. The *BEAM* function was used to identify genes that are differentially expressed between the three main branches of the trajectory (qval <1e-20). Code available upon request.

## Numerical model

Codes generated for the numerical model are available at: http://doi.org/10.17632/j2j9gmyb6m.1.

### 1. Algorithm of the clone model

In this section we detail the stochastic numerical model used to generate clones.

Each clone grows from a single initial cell. There are two cell types, Chinmo⁺Imp⁺ tNBs (which we will call C cells for simplicity) and Syp⁺E93⁺ tNBs (called S cells). The probability that the initial cell is a C cell is $p_c$ and the probability that the initial cell is a S cell is $p_s = 1 - p_c$.

We assume that C and S cells have constant division rates, $k_c$ and $k_s$ respectively. We define the division times as the inverses of the division rates, so that $T_c = 1/k_c$ and $T_s = 1/k_s$. An implicit assumption is that we neglect any refractory period following a division. Divisions are considered as random events occurring at a certain rate.

We use T = 1 hr long time steps. The growth of a model clone during 10 days thus runs over 240 time steps.

At each time step, for each C cell (resp. S cell) in the clone, the probability to undergo division is equal to $\int_0^T \frac{1}{T_c} e^{-t/T_c} \, dt = 1 - e^{-T/T_c}$ (resp. $1 - e^{-T/T_s}$).

If a C cell undergoes division, it can either:

- Generate 2 C cells, with probability $P(c \rightarrow cc)$
- Generate 1 C cell and 1 S cell, with probability $P(c \rightarrow cs)$
- Generate 2 S cells, with probability $P(c \rightarrow ss) = 1 - P(c \rightarrow cc) - P(c \rightarrow cs)$

Similarly, if a S cell undergoes division, it can either:

- Generate 2 S cells, with probability $P(s \rightarrow ss)$
- Generate 1 C cell and 1 S cell, with probability $P(s \rightarrow cs)$
- Generate 2 C cells, with probability $P(s \rightarrow cc) = 1 - P(s \rightarrow ss) - P(s \rightarrow cs)$

In addition, each newly created C cell (resp. S cell) has a probability $q_c$ (resp. $q_s$) to be quiescent. Note that this also applies to the initial cell of the clone. Quiescent cells lose their ability to undergo further divisions. In this simplified model, this probability does not depend on cell generation (as cells have no age in the model). However, it is likely that "older" cells have a higher chance to become quiescent. This is not taken into account here.

To produce plots of size distributions and of composition ratios, we simply generate a large number of clones (N = 1000) by iterating this algorithm. Due to the stochastic nature of the algorithm, there is important size and composition variability among simulated clones, which we can compare to the variability observed in vivo.

### 2. Reducing the amount of independent parameters

This general model has 9 independent parameters: $p_c$, $T_c$, $T_s$, $P(c \rightarrow cc)$, $P(c \rightarrow cs)$, $P(s \rightarrow ss)$, $P(s \rightarrow cs)$, $q_c$ and $q_s$.

To reduce the number of independent parameters, we performed experimental measurements in clones that allow to establish quantitative relationships between the parameters.

#### a. Two-cell clones

First, we analyzed clones composed of exactly two cells. To that end, we chose to look tumors 2 days after clonal induction, as at this stage many clones are composed of two cells (*Figure 4—figure supplement 1*).

The probability for a two-cell clone to be composed of two C cells is:

$$P(CC) = p_c \, P(c \rightarrow cc) + p_s \, P(s \rightarrow cc) \tag{1}$$

Similarly,

$$P(CS) = p_c \, P(c \rightarrow cs) + p_s \, P(s \rightarrow cs) \tag{2}$$

and

$$P(SS) = p_c \, P(c \rightarrow ss) + p_s \, P(s \rightarrow ss) = 1 - P(CC) - P(CS) \tag{3}$$

We estimated $P(CC)$, $P(CS)$ and $P(SS)$ from N=122 2-cell clones (7 animals), and found that $P(CC) \approx 0.29$, $P(CS) \approx 0.07$ and $P(SS) \approx 0.64$.

Using these values in *Equations (1) and (2)* reduces the number of independent parameters from 9 to 7 (as *Equation (3)* is not independent from *Equations (1) and (2)*).

### b. Quiescent cells

We analyzed clones 8 days ACI. We reasoned that clones still composed of a single cell at this stage were most likely initiated from an initially quiescent cell. Hence the probability $P(QC)$ to observe a clone composed of a single C cell at 8 days is equal to the probability for an initial cell to be a quiescent C cell, so that:

$$P(QC) = p_c \, q_c \tag{4}$$

Similarly, the probability that a clone is composed of a single quiescent S cell is

$$P(QS) = p_s \, q_s \tag{5}$$

We estimated $P(QC)$ and $P(QS)$ from N=338 clones 8 days after induction (5 animals), and found that $P(QC) \approx 0.02$ and $(QS) \approx 0.26$. Using these values in equations (4) and (5) reduces the number of independent parameters by another 2.

### c. Hierarchical scenario

We are left with 5 independent parameters. We showed in the main text that experimental clone data has the signature of a hierarchical growth mode (maintenance of clones composed of S cells only). In the following we thus rule out the generation of C cells by S cells. Hence $P(s \rightarrow ss) = 1$, and $P(s \rightarrow cs) = P(s \rightarrow cc) = 0$.

In addition, if C cells can only be generated by C cells, we can estimate the division time $T_c$ of C cells. At time t, the number of cells in clones only composed of C cells should be on average equal to $N_c(t) = e^{t/T_c}$.

At 2 days, we find that clones composed only of C cells are composed, on average, of 3.4 cells (N=75 clones from 5 animals). Hence $T_c \approx \frac{2}{\ln(3.4)} = 1.6$ days.

Hence, we are left with only two independent parameters, from which all the others can be determined. We arbitrarily chose $T_s$ and $P(c \rightarrow cc)$ as these two parameters, and used these as free parameters to perform the fits.

## 3. Fitting experimental data

To determine the values of $T_s$ and $P(c \rightarrow cc)$, we adopted a fitting approach. We want to determine the values of $T_s$ and $P(c \rightarrow cc)$ that fit best our experimental data. To that end we generated error maps in the $(P(c \rightarrow cc), T_s)$ space (*Figure 4—figure supplement 2*). To generate the error maps, we computed 1000 clones for each pixel (100x100) on the map.

We calculate the error on size as the Kolmogorov-Smirnov distance between model and experimental clone size distributions, for all clone categories and at all stages (12 terms).

We calculate the error on composition as the normalized sum of Euclidian distances between model and experimental proportions of each clone category at all stages (12 terms).

The combined error is the average between the normalized error on size and the normalized error on composition.

We find that the smallest error is found for $P(c \rightarrow cc) \approx 0.64$ and $T_s \approx 1.3$ days, and thus (*Equations 1-5*):

$$p_c \approx 0.45$$

$$P(c \rightarrow cs) \approx 0.15$$

$$P(c \rightarrow ss) \approx 0.21$$

$$q_c \approx 0.04$$

$$q_s \approx 0.47$$

We use these values to generate *Figures 4C–G* and *5A,B*, and *Figure 4—figure supplement 2*.

## 4. Tumor composition and deterministic version of the model
### a. Proportion of S and C cells in tumors

To test whether our model also accounts for the stabilization of the volume fraction occupied by S and C cells in tumor, we simulated the growth of clones on longer periods (30 days), and plotted the overall proportion of each cell type as a function of time (*Figure 5A*).

### b. Deterministic, continuous model

To better understand this homeostatic behavior without systematically simulating thousands of clones, we wrote a deterministic version of the model where the exact same ingredients and parameters are used.

For simplicity, we call $C$ the number of C cells, $S$ the number of S cells, $C_q$ the number of quiescent C cells and $S_q$ the number of quiescent S cells. The rate of change of $C$ writes:

$$\frac{dC}{dt} = -k_c C + (1-q_c)(2k_{c\to cc} + k_{c\to cs})C + (1-q_c)(2k_{s\to cc} + k_{s\to cs})S \tag{6}$$

With $k_{x\to yz} = k_x P(x \to yz)$, where $x$, $y$ and $z$ are cell types (C or S).

Each C cell division removes 1 C cell (first term). C cell divisions can also generate 1 or 2 C cells with given rates (second term), with a possibility that they are quiescent (prefactor $(1-q_c)$). S cell divisions can also generate C cells (third term), which can also be quiescent (prefactor).

Similarly, we can write the rate of change of $S$:

$$\frac{dS}{dt} = -k_s S + (1-q_s)(2k_{s\to ss} + k_{s\to cs})S + (1-q_s)(2k_{c\to ss} + k_{c\to cs})C \tag{7}$$

And for quiescent cells:

$$\frac{dC_q}{dt} = q_c(2k_{c\to cc} + k_{c\to cs})C + q_c(2k_{s\to cc} + k_{s\to cs})S \tag{8}$$

$$\frac{dS_q}{dt} = q_s(2k_{s\to ss} + k_{s\to cs})S + q_s(2k_{c\to ss} + k_{c\to cs})C \tag{9}$$

Note that this continuous model is fully deterministic, and therefore not suited for the simulation of clone size distribution and composition, which are inherent to the stochastic aspects of divisions. However, it can recapitulate the overall proportion of cell types in a large population of cells. Integrating this system of equations, we thus computed the ratios $\frac{C+C_q}{N_{cells}}$ and $\frac{S+S_q}{N_{cells}}$, that is, the respective proportions of C and S cells. We first compared the outcome of the deterministic model to the outcome of the clone model using the exact same parameters. We find that the deterministic model perfectly fits the stochastic model (*Figure 5A*), and reaches the same homeostatic state. Importantly, this is also a control that both algorithms were properly implemented. We also use this deterministic model to illustrate the influence of the initial proportions of C and S cells (*Figure 5B*).

## Acknowledgements

We thank I Davis, P Macdonald, DJ McKay, C Meignin M Noll, N Sokol for flies and antibodies. We are grateful to B Aigouy for helping with the 'Tissue Analyser' plugin, C Hérouard-Molina for performing cDNA libraries and sequencing at the IGBMC GenomEast platform, A Saurin for alignment of RNA-seq data, P Outters at HalioDx (Marseille) for generating the single-cell transcriptomic data, F Hubert and J Oliver for discussions on mathematical modeling, JC Patel for help with image analysis and P Grenot at Ciphe for help with FACS. We also acknowledge the Bloomington Drosophila Stock Center (NIH P40OD018537), the Vienna Drosophila RNAi Center (VDRC), TRiP at Harvard Medical School (NIH/NIGMS R01-GM084947), Kyoto DGRC and NIG-Fly Stock Centers for flies, and the Developmental Studies Hybridoma Bank (DSHB) for monoclonal antibodies. Bulk tumor

sequencing was performed by the IGBMC GenomEast platform, a member of the 'France Génomique' consortium (ANR-10-INBS-0009)". We thank France-BioImaging/PICsL infrastructure (ANR-10-INSB-04–01). We thank S Kerridge for critical reading of the manuscript.

## Additional information

### Funding

| Funder | Grant reference number | Author |
|---|---|---|
| Fondation ARC pour la Recherche sur le Cancer | PJA20141201621 | Cédric Maurange |
| Fondation ARC pour la Recherche sur le Cancer | Fin de thèse | Sara Genovese |
| Canceropôle PACA | Starting Package Single Cell | Cédric Maurange |
| Centre National de la Recherche Scientifique | | Cédric Maurange |
| Aix-Marseille Université | Bio trail | Sara Genovese |
| Ligue Contre le Cancer | Comprendre les principes fondamentaux régissant la hiérarchie cellulaire dans les tumeurs neurales | Cédric Maurange |

The funders had no role in study design, data collection and interpretation, or the decision to submit the work for publication.

### Author contributions

Sara Genovese, Conceptualization, Formal analysis, Visualization, Methodology, Writing—original draft, Writing—review and editing; Raphaël Clément, Conceptualization, Formal analysis, Investigation, Visualization, Methodology, Writing—original draft, Writing—review and editing; Cassandra Gaultier, Conceptualization, Data curation, Formal analysis, Investigation, Visualization, Methodology, Writing—original draft, Writing—review and editing; Florence Besse, Conceptualization, Formal analysis, Visualization, Methodology, Writing—review and editing; Karine Narbonne-Reveau, Conceptualization, Data curation, Formal analysis, Supervision, Funding acquisition, Validation, Investigation, Visualization, Writing—original draft, Writing—review and editing; Fabrice Daian, Data curation, Visualization; Sophie Foppolo, Methodology, Project administration; Nuno Miguel Luis, Investigation, Methodology, Writing—original draft, Writing—review and editing; Cédric Maurange, Conceptualization, Formal analysis, Supervision, Funding acquisition, Investigation, Visualization, Writing—original draft, Writing—review and editing

### Author ORCIDs

Cassandra Gaultier http://orcid.org/0000-0001-6401-9163
Florence Besse http://orcid.org/0000-0003-4672-1068
Nuno Miguel Luis http://orcid.org/0000-0001-5438-9638
Cédric Maurange https://orcid.org/0000-0001-8931-1419

### Decision letter and Author response

Decision letter https://doi.org/10.7554/eLife.50375.034
Author response https://doi.org/10.7554/eLife.50375.035

## Additional files

### Supplementary files

• Supplementary file 1. Top200 genes dynamically regulated along the pseudotime.
DOI: https://doi.org/10.7554/eLife.50375.024

• Supplementary file 2. Differentially expressed genes between $pox^n > pros^{RNAi}$, $Syp^{RNAi}$ tumors and $pox^n > pros^{RNAi}$ tumors.
DOI: https://doi.org/10.7554/eLife.50375.025

• Supplementary file 3. Heatmap depicting dynamically expressed genes along the different branches of the trajectory in $pox^n > pros^{RNAi}$ tumors. Prebranch starts at the root of the trajectory. Cell fate one corresponds to the $E93^+stg^+$ branch. Cell fate two corresponds to the $E93^+E(spl)^+$ branch as labeled in *Figure 7A*.
DOI: https://doi.org/10.7554/eLife.50375.026

• Transparent reporting form
DOI: https://doi.org/10.7554/eLife.50375.027

### Data availability

Sequencing data have been deposited in GEO under accession codes GSE114986 and GSE114562. Codes generated for the numerical model are available at: http://doi.org/10.17632/j2j9gmyb6m.1. Codes used for single-cell RNA-seq analysis are available at: https://github.com/cedricmaurange/Genovese-et-al.-2019 (copy archived at https://github.com/elifesciences-publications/Genovese-et-al.-2019).

The following datasets were generated:

| Author(s) | Year | Dataset title | Dataset URL | Database and Identifier |
| --- | --- | --- | --- | --- |
| Gaultier C, Daian F, Maurange C | 2018 | Single-cell RNA-seq analysis of Drosophila neuroblast tumors | https://www.ncbi.nlm.nih.gov/geo/query/acc.cgi?acc=GSE114986 | NCBI Gene Expression Omnibus, GSE114986 |
| Genovese S, Clément R, Gaultier C, Besse F, Narbonne-Reveau K, Daian F, Foppolo S, Miguel Luis N, Maurange C | 2018 | RNA-binding proteins govern neural progenitor hierarchy during development and tumorigenesis | https://www.ncbi.nlm.nih.gov/geo/query/acc.cgi?acc=GSE114562 | NCBI Gene Expression Omnibus, GSE114562 |

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
