## [Decision Letter]

Thank you for submitting your work entitled "Cooption of antagonistic RNA-binding proteins establishes cell hierarchy in *Drosophila* neuro-developmental tumors" for consideration by *eLife*. Your article has been reviewed by two peer reviewers, and the evaluation has been overseen by a Reviewing Editor and a Senior Editor. The following individual involved in review of your submission has agreed to reveal their identity: Bart Deplancke (Reviewer #1).

Our decision has been reached after consultation between the reviewers. Based on these discussions and the individual reviews below, we regret to inform you that your work will not be considered further for publication in *eLife* at this point as there are a number of additional experiments needed that would likely take more than two months to complete. However, if you are able to address all the reviewers' concerns and wish to resubmit a revised article in the future, we would be happy to consider this.

The most important revision points are listed first but please address all the other points raised in the reviewers' reports too.

Essential Revisions:

1) The ubiquitous expression of Chinmo and Syp in all identified clusters possess a problem that weakens the model. Resolving the following issues is fundamental. As it is presented now, the model is not supported by the data.

While Chinmo and Syp are key genes to mark the early and late NBs, the scRNA data shows they are rather universally expressed in all clusters This may be under translational control, which is unfortunately not explored further despite being key for the model that they present in the Discussion.

Similarly, the binding of Imp and Syp to chinmo UTRs implies regulation at the RNA level, but chinmo and syp are both ubiquitously expressed. Here, post-transcriptional regulation seems unlikely given the striking mRNA expression overlap between chinmo and syp.

2) The authors indicate that Imp and Chinmo "form a positive feedback loop", which implies a high extent of co-expression, yet the scRNA-seq data shows that imp is expressed in a minor fraction of chinmo positive cells.

3) The concept of stable heterogeneity needs further experimental substantiation by, for instance, determining if the ratio/behaviour of cell types found in adult porsRNAi brain persists in implanted tumours. Linked to clonal analysis with flybow in the transplanted tumour the results from this experiment would be very informative about the validity of the model.

4) The states with high expression of Imp were put at the beginning of the pseudotime using Monocle. However, Monocle cannot be used to determine the root or the base state of the trajectory. Use RNA velocity to provide further substantiation of the pseudotime ordering.

5) Tumour tissue segmentation using Tissue Analyser must be validated by providing high quality images and their corresponding segmented processed images illustrating that (i) the Mira signal is recognised as a "membrane reference",(ii) the images are segmented accurately with respect to nuclei number, and (iii) cell size distribution is accurately generated.

6) Provide evidence showing that mixed clones result from a common precursor (as argued) rather than from cells moving around or from Chinmo+ cells turning down Chinmo protein levels and upregulating Syp.

7) Provide substantiating evidence when referring to asymmetric differentiation divisions.

8) Most of the analyses focuses on Chinmo, Imp, and Syp, which were already previously implicated in this process. Beyond providing a view on tumor heterogeneity, the scRNA-seq data remains underexplored with respect to new insights into tumor development. Related to this, the model prediction of growth being mostly due fast proliferation of Chinmo+ cells appears somewhat trivial. Please, make a clear statement of what we learned from this exercise.

9) Does experimental evidence derived from only one time-point substantiate the claims on "persistence"?

10) There is a long list of critical quality control data that must be provided (i.e. mapping rate of the sequencing reads; percentage of mitochondria and rRNA reads; scRNA-seq data normalization; clusters stability and others).

Reviewer #1:

Genovese et al., present a set of experiments investigating the hierarchical regulation of neural tumours stemming from asymmetrically-dividing neural progenitors (neuroblasts, NB) in *Drosophila melanogaster*, mainly focusing on a pair of established RNA-binding proteins Imp and Syncrip, whose antagonistic activity regulates self-renewing potential within the studied system.

The presented study encompasses an elegantly designed and executed collection of assays, which in a relatively comprehensive manner address the posed questions. These begin with the single-cell RNA sequencing experiment, and as much as the authors do not seem to explore the abundance and more global relevance of the obtained results to the fullest, they provide a thorough validation of NB tumours (tNB) development and hierarchical regulation- related phenomena. This validation includes refined clonal analysis and stochastic modelling of the hierarchical scheme of tumour development followed by quantitative analysis of tumour growth. This is completed by an in-depth molecular dissection of the dynamics of tNB development and progression upon silencing or overexpression of relevant regulatory actors as well as elegant in situ immunofluorescence assays incorporating analysis of the number, occurrence frequency and size of tNBs at different developmental stages. Finally, they interrogate the metabolic changes within the tumours by integrating their transcriptomic and molecular data sets.

In sum, this is a well-crafted study with a compelling message. However, several concerns need to be addressed prior to publication:

a) The authors started their study with an scRNA-seq analysis on tNBs. The resulting data are potentially a valuable resource, however, the description of the data is rather crude and its integration in the rest of the paper rather shallow (see also below). This is best illustrated by the fact that most downstream analyses focus on Chinmo, Imp, and Syp, genes that were already previously implicated in this process. Beyond providing a view on tumor heterogeneity, the scRNA-seq remain therefore somewhat underexplored with respect to providing novel molecular / regulatory insights regarding tumor development. Below, a few recommendations:

Technical:

• Quality control. Expect for indicating that "we sequenced 5796 cells with median number of 1806 genes/cell", the author did not provide any other QC measures. What is the mapping rate of the sequencing reads? What is the percentage of mitochondria and rRNA reads? <2,000 genes is rather low, what could be the reason for this? How many reads and UMI per cells were acquired?

• How are the scRNA-seq data normalized. Was any filtering performed? If yes, which criteria were used?

• How was the tSNE map plotted? Is it based on the most variable genes? If so, how were the most variable genes determined?

• The authors mention in the figure legend that "unsupervised clustering using the.… of the Cell Ranger". However, detailed parameters were not provided. Additionally, what are the differentially expressed genes in the clusters, especially cluster 7, onto which the entire study is based.

• As a sanity check, the authors should plot the expression level of pros in the scRNA-seq data.

• How stable are the clusters? Was any silhouette analysis performed?

Biological:

• While Chinmo and Syp are key genes to mark the early and late NBs, the scRNA data shows they are rather universally expressed in all identified clusters (Figure 1B, F). As the authors pointed out (e.g. Figure 1G), they may be under translational control, which is unfortunately not explored further despite being key for the model that they present in the Discussion. For example, the authors indicate that Imp and Chinmo "form a positive feedback loop", which implies a high extent of co-expression, yet the scRNA-seq data shows that imp is expressed in a minor fraction of chinmo positive cells. Similarly, the authors show that Imp and Syp both bind chinmo UTRs, implying (because not elaborated further) regulation at the RNA level with perhaps Imp having a stabilizing function and Syp a destabilizing one. But how can the authors then explain that chinmo and syp are both ubiquitously expressed? Again, one can evoke post-transcriptional regulation here, but given the striking mRNA expression overlap between chinmo and syp, such regulation (and thus large disconnect between mRNA and protein levels) would be rather striking. Resolving these questions seems fundamental to this study since, as it is presented now, the model is not supported by the presented data.

• The authors use Monocle to analyze the trajectory of tNBs, and find that "the states with highly expression of Imp were put at the beginning of the pseudotime". However, while Monocle is designed to find the transition trajectory/pseudotime based on transcriptome ordering, it cannot be used to determine the root or the base state of the trajectory. In addition, it is becoming common practice to seek independent analytical validation for such pseudotime analyses and in this regard, the "RNA velocity" tool (https://github.com/velocyto-team/velocyto.R) is a very exciting and intuitive approach. Finally, it would be valuable if staining of Chimo/Imp/Syp/E93 in one microscopy field would be performed to show that Chimo/Imp and Syp/E93 truly represent two different NB stages.

• The authors focus on cluster 7 and functionally validate its biological relevance. While functionally characterizing all clusters are beyond the scope of this manuscript, the author should at least discuss their significance. The cluster at the right side (Figure 1C, maybe cluster 1 or 8) for example is much separated from the other clusters. Which cells belong to this cluster? Any enrichment for specific biological processes? What can we learn here?

Reviewer #2:

The key question this study tries to answer is how cellular heterogeneity comes about in human neural tumors using *Drosophila* as a model system. Overall there are a lot of interesting observations in the manuscript, but the main weakness is that it is unclear whether the model system used is suitable to address the question. The manuscript suffers further from extensive borrowing of concepts and terminology that are often used in a very confusing and misleading manner and the paper is written in a highly convoluted way, which makes it a rather tough read. Perhaps the story tries to look at the problem from too many angles, many of them remain largely descriptive (e.g. single cell RNA seq, the modeling, the bulk tumor sequencing). Human tumor heterogeneity is apparent in complex factettes. One important idea is the co-existence of genetically divergent tumor cell clones and many concepts are around to explain their origin, aspects of deregulated development certainly contribute. This manuscripts addresses this later aspect. One could look at the data, however, by judging the analysed tissue as a relatively normal tissue in which certain aspects of normal neurogenesis are due to the applied manipulations not functional, which on its own right is interesting, but framing the findings in the context of the above question appears forced. For instance, many of the molecular mechanism already identified can be directly applied. This contrasts with the observation that in other malignant tumors, that can be experimentally induced, genome instability arises rapidly, if that were the case here, which we don't know, doing genetics (flybow) and interpreting the results becomes much more complex. If genetics works predictably, as for instance the clonal analysis suggests, then this casts doubt on the tumor-like state of the tissue analysed. Looking at the data from a developmental biology point of view, the findings may rather represent faulty neurogenesis in which decommissioning of neural stem cells and interfering with the temporal transcription factor cascades has not occur properly, but that it appears that the cells in question can responds to the normal constraints of fly development in as much as it can when a key regulatory factor is removed. Removing more factors alters/worsens of course the situation.

However, there are certainly valuable insights into fly specific developmental biology processes of neurogenesis and decommissioning of neuroblast divisions.

---

## [Author Response]

Essential Revisions:1) The ubiquitous expression of Chinmo and Syp in all identified clusters possess a problem that weakens the model. Resolving the following issues is fundamental. As it is presented now, the model is not supported by the data.While Chinmo and Syp are key genes to mark the early and late NBs, the scRNA data shows they are rather universally expressed in all clusters This may be under translational control, which is unfortunately not explored further despite being key for the model that they present in the Discussion.Similarly, the binding of Imp and Syp to chinmo UTRs implies regulation at the RNA level, but chinmo and syp are both ubiquitously expressed. Here, post-transcriptional regulation seems unlikely given the striking mRNA expression overlap between chinmo and syp.

The fact that chinmo mRNA is detected in most if not all tNBs is not surprising given that it is known to be regulated post-transcriptionally in NBs during development (Zhu et al., 2006; Liu et al., 2015; Dillard et al., 2018). Here, we clearly demonstrate that it is also regulated post-transcriptionally in tumors. First, because the Chinmo protein is only present in cells also expressing Imp, as shown by immunostainings in Figure 1E. Second, because when mis-expressing, in all tNBs, a UAS-transgene containing the coding sequence of mCherry flanked by the UTRs of the chinmo mRNA, we observe that the mCherry protein is only present in tNBs also expressing the endogenous Chinmo protein (new Figure 1—figure supplement 4). This demonstrates that regulation via the chinmo UTRs can recapitulate the expression pattern of endogenous Chinmo in tumors.

Intriguingly, Syp has been shown to be mainly regulated at the transcriptional level in NBs during development. In contrast, we clearly find that it is mainly regulated at the post-transcriptional level in tumors since Syp mRNA is detected in most tNBs while Syp protein is only present in Chinmo-negative tNBs (Figure 1D). This is thus an aspect of Syp regulation that appears to be tumor-specific and that we do not clearly understand. Consistently, we know that Syp is hardly detectable by immunostaining upon mis-expression of a UAS-Syp transgene in Chinmo^+^Imp^+^ NBs both during development and tumorigenesis (data not shown). This would be consistent with a post-transcriptional repression of Syp in Chinmo^+^Imp^+^ tNBs.

In essence, we do not really understand the mechanisms that post-transcriptionally silence Syp in Chinmo+Imp+ tNBs, but we do not think that this undermines our current model as our genetic experiments demonstrates the essential role of Syp in silencing chinmo in tumors. This aspect of chinmo and Syp post-transcriptional regulation is now more substantially discussed in the first paragraph of the Results section.

2) The authors indicate that Imp and Chinmo "form a positive feedback loop", which implies a high extent of co-expression, yet the scRNA-seq data shows that imp is expressed in a minor fraction of chinmo positive cells.

This positive feedback loop was described in our previous publication (Narbonne-reveau et al., 2016). In this loop, Chinmo promotes Imp expression at the transcriptional level while Imp promotes Chinmo expression at the post-transcriptional level. A consequence of this transcriptional/translational cross-regulatory interaction is that Chinmo protein can only be produced in tNBs expressing the Imp protein, while *Imp* can only be transcribed in tNBs expressing the Chinmo protein. Thus, even if chinmo mRNA is ubiquitous, the positive feedback loop is only functioning in the fraction of tNBs expressing the Imp protein. This feedback loop provides robustness to perpetuate the population of Chinmo^+^Imp^+^ tNBs that acts as cancer stem cells, as shown later in the manuscript.

3) The concept of stable heterogeneity needs further experimental substantiation by, for instance, determining if the ratio/behaviour of cell types found in adult porsRNAi brain persists in implanted tumours. Linked to clonal analysis with flybow in the transplanted tumour the results from this experiment would be very informative about the validity of the model.

We agree that this would be a very interesting experiment to perform. However, transplantation implies major changes from the initial microenvironment of the primary tumor and different types of stresses induced by cuts and wounding responses in the tumor for example… In our opinion, this is very likely to interfere with the tumor-intrinsic mechanisms that regulate the population of Chinmo+Imp+ cancer stem cell like-cells in the primary tumor. Therefore, the interpretation of this transplantation experiment might be complex, and the results may not be applicable to the primary tumor.

Although we did not perform transplantations for the aforementioned reasons, we nevertheless consolidated our claim that tumor heterogeneity evolves towards an equilibrium from larval stages to adulthood by measuring the proportions of the two major tNB types (Chinmo+Imp+ and Syp+E93+) at different time points during adult stages (now at d8, d10, d11 and d15 after tumor induction, respectively corresponding to d1, d3, d4 and d8 into adulthood) (Figure 2E). For every time points, we find that the proportion of Chinmo+Imp+ tNBs remains approximately stable (around 20% +/- 10%).

4) The states with high expression of Imp were put at the beginning of the pseudotime using Monocle. However, Monocle cannot be used to determine the root or the base state of the trajectory. Use RNA velocity to provide further substantiation of the pseudotime ordering.

We agree with the reviewer. In order to circumvent this problem, we made use of our clonal analysis/modeling approach that unambiguously demonstrates that Imp+ tNBs are at the top of the hierarchy in tumors (Figure 4). This allowed us to perform a semi-supervised pseudotime ordering using Monocle where tNBs expressing Imp where put at the root of the pseudotime. Since the pseudotime analysis now relies on the clonal and numerical analysis, it has now been moved to Figure 7. We think that this reorganization of the manuscript is more logic and reinforces the value of the clonal analysis and modelling.

5) Tumour tissue segmentation using Tissue Analyser must be validated by providing high quality images and their corresponding segmented processed images illustrating that (i) the Mira signal is recognised as a "membrane reference",(ii) the images are segmented accurately with respect to nuclei number, and (iii) cell size distribution is accurately generated.

We have added a new figure of better quality (Figure 8—figure supplement 3) with the mira staining used as a membrane reference. The segmentation was double-checked manually using the dapi staining for nuclear reference.

6) Provide evidence showing that mixed clones result from a common precursor (as argued) rather than from cells moving around or from Chinmo+ cells turning down Chinmo protein levels and upregulating Syp.

As for the possibility that cells from different clones may migrate and mix to generate mixed clones, we have now added a new picture showing a large clone stained 12 days after random induction at low frequency (Figure 3—figure supplement 1). As observed, the fact that the cells composing the clones remain grouped and do not disperse suggests that tNBs, in general, do not actively migrate although rearrangements with neighboring cells probably exist. We have also measured †that between 1 or 2% of cells undergo the mitotic recombination allowing mCherry expression in tumors. Therefore, the probability that two neighboring cells undergo a recombination event is rather rare. Together, these observations suggest that mixed clones are unlikely to results from tNBs originating from different clones and mixing together.

Having excluded this possibility, our numerical simulations suggests that mixed clones can only result from an initial Chinmo+Imp+ tNB that has divided to give rise to populations of Chinmo+Imp+ and populations of Syp+E93+ tNBs.

7) Provide substantiating evidence when referring to asymmetric differentiation divisions.

We now provide a new figure (Figure 4—figure supplement 1) with three examples of two-cell clones.

1) The two cells are chinmo+.

2) One cell is Chinmo+ the other is Chinmo-

3) The two cells are chinmo-

“Example 1” is likely to result from the symmetric self-renewing division of a Chinmo^+^Imp^+^ tNB.

“Example 2” could result from a fate asymmetric division of a Chinmo^+^Imp^+^ tNB.

“Example 3” may result from either a symmetric differentiating division of a Chinmo^+^Imp^+^ tNB or from a symmetric self-renewing division of a Syp^+^E93^+^ tNB.

8) Most of the analyses focuses on Chinmo, Imp, and Syp, which were already previously implicated in this process. Beyond providing a view on tumor heterogeneity, the scRNA-seq data remains underexplored with respect to new insights into tumor development. Related to this, the model prediction of growth being mostly due fast proliferation of Chinmo+ cells appears somewhat trivial. Please, make a clear statement of what we learned from this exercise.

We have now generated a much more extensive analysis of our scRNA-seq data, comparing our data in tumors with recent work performed in larval NBs to show that only a subset of the genes previously shown to be temporally regulated in larval NBs are also dynamically regulated in tumors, substantiating the fact that a large part of the temporal patterning system, other than Chinmo, Imp and Syp) is redeployed in tumors.

We have also performed a more thorough pseudotime analysis to consolidate our claims that glucose metabolism genes are down-regulated along the pseudotime, together with cell-cycle genes (Figure 7 and 8 and Figure 8—figure supplement 2:). This demonstrates that glucose metabolism genes during tumorigenesis are regulated by the coopted temporal patterning system.

Concerning what we can learn from the clonal and modelling approach:

1) The model uses the clonal analysis to demonstrate that tumor growth is governed by a hierarchical organization. This knowledge is important if one want, in the future, understand the rules that regulate the balance between CSC self-renewal and differentiation and therefore the proportion of CSCs in the tumor.

2) Thanks to the model, we could determine which cells compose the root of the pseudotime.

3) The model also resolves the paradoxical observation that Chinmo^+^Imp^+^ tNBs are in minority compared to Syp^+^E93^+^ tNBs (shown in Figure 2E), while exhibiting a higher average mitotic index than Syp^+^E93^+^ tNBs (shown in Figure 2D).

4) We find it remarkable that the model can predict a division scheme that can recapitulate both clonal growth (Figure 4D) and the dynamics of tumor heterogeneity along a period of 15 to 20 days (Figure 5), before adult flies are killed by the tumor. Although, like all models, we don’t expect our model to perfectly recapitulate all the parameters regulating tumor growth and heterogeneity (for example, we have neglected apoptosis and neuronal differentiation that occur at low levels), we think it provides a reasonable and useful ground on which further studies can be performed for a more detailed understanding.

5) The model predicts that there are 2 types of Syp+E93+ tNBs: a proliferative type and a quiescent type. Substantiating this prediction, we now find using Monocle that tNBs follow a differentiation trajectory that splits in two within the population of Syp+E93+ tNBs. One population keep expressing medium levels of glucose metabolism genes and cell-cycle genes, including high levels of the CDC25 gene string (stg). We now refer to this population as E93+stg+ tNBs. The second population express low levels of glucose metabolism genes, cell cycle genes but high levels of E(spl) genes. We therefore propose that this population of E93+E(spl)+ tNBs are quiescent. Interestingly, in vertebrates, the orthologous Hes genes are highly expressed in quiescent neural stem cells.

In conclusion, we believe that our clonal and modeling approach allowed us to make the first firm demonstration that hierarchical tumors exist in *Drosophila*, thus providing a new paradigm for cancer research.

More importantly, by uncovering a network of temporal patterning genes that is responsible:

i) for “transforming” a pool of proliferative neural stem cells into a hierarchical tumor,

ii) for governing the hierarchical, heterogeneity and metabolic properties of tumor cells, we demonstrate an overarching role of temporal patterning genes in the initiation and progression of neural tumors with an early developmental origin. Given the recent finding that temporal patterning is conserved in mammals, our work could have wide implications for understanding how pediatric neural cancers develop so rapidly despite often carrying very few mutations.

Moreover, we know describe many new genes associated with a cancer stem cell identity. Investigating their function will form the basis for future studies aiming at understanding how the balance between cancer stem cell self-renewal and differentiation is regulated in tumors.

9) Does experimental evidence derived from only one time-point substantiate the claims on "persistence"?

I think this question refers to the experiment described in the now Figure 6—figure supplement 1, where we inactivated Syp in larval NBs. This leads to the robust persistence of NBs in pharate adults. This “persistence” phenotype has been described in more details in a recent manuscript (Yang et al. 2017), therefore it was not our intention to repeat at different time points. Instead, we use this experiment to decipher the underlying mechanisms of persistence, and demonstrate that the positive feedback loop operating between chinmo and Imp is responsible for NB persistence in the SypKD context. These results demonstrate the key role of the chinmo/Imp module in promoting a default self-renewing state that is particularly active during early development.

10) There is a long list of critical quality control data that must be provided (i.e. mapping rate of the sequencing reads; percentage of mitochondria and rRNA reads; scRNA-seq data normalization; clusters stability and others).

The data were completely reanalyzed using the standard procedure for Seurat*v3.0.2* described in https://satijalab.org/seurat/v3.0/pbmc3k_tutorial.html for clustering. Details are now given in the new Materials and methods section. This has led to a new Figure 1, Figure 1—figure supplement 1 and 2. QC metrics such as the number of UMI/cells, genes/cell and percentage of mitochondrial RNA are now shown. The data we have obtained are standard for *Drosophila* cells when comparing with other publications (Davie et al., 2018; PMID: 29909982).

The pseudotime analysis was also repeated with a semi-supervised procedure as described in http://cole-trapnell-lab.github.io/monocle-release/docs/#constructing-single-cell-trajectories (Figure 7).

Reviewer #1:[…] In sum, this is a well-crafted study with a compelling message. However, several concerns need to be addressed prior to publication:a) The authors started their study with an scRNA-seq analysis on tNBs. The resulting data are potentially a valuable resource, however, the description of the data is rather crude and its integration in the rest of the paper rather shallow (see also below). This is best illustrated by the fact that most downstream analyses focus on Chinmo, Imp, and Syp, genes that were already previously implicated in this process. Beyond providing a view on tumor heterogeneity, the scRNA-seq remain therefore somewhat underexplored with respect to providing novel molecular / regulatory insights regarding tumor development. Below, a few recommendations:Technical:• Quality control. Expect for indicating that "we sequenced 5796 cells with median number of 1806 genes/cell", the author did not provide any other QC measures. What is the mapping rate of the sequencing reads? What is the percentage of mitochondria and rRNA reads? <2,000 genes is rather low, what could be the reason for this? How many reads and UMI per cells were acquired?• How are the scRNA-seq data normalized. Was any filtering performed? If yes, which criteria were used?• How was the tSNE map plotted? Is it based on the most variable genes? If so, how were the most variable genes determined?• The authors mention in the figure legend that "unsupervised clustering using the.… of the Cell Ranger". However, detailed parameters were not provided. Additionally, what are the differentially expressed genes in the clusters, especially cluster 7, onto which the entire study is based.• As a sanity check, the authors should plot the expression level of pros in the scRNA-seq data.• How stable are the clusters? Was any silhouette analysis performed?

Please look above at our detailed response to point 10.

Biological:• While Chinmo and Syp are key genes to mark the early and late NBs, the scRNA data shows they are rather universally expressed in all identified clusters (Figure 1B, F). As the authors pointed out (e.g. Figure 1G), they may be under translational control, which is unfortunately not explored further despite being key for the model that they present in the Discussion. For example, the authors indicate that Imp and Chinmo "form a positive feedback loop", which implies a high extent of co-expression, yet the scRNA-seq data shows that imp is expressed in a minor fraction of chinmo positive cells. Similarly, the authors show that Imp and Syp both bind chinmo UTRs, implying (because not elaborated further) regulation at the RNA level with perhaps Imp having a stabilizing function and Syp a destabilizing one. But how can the authors then explain that chinmo and syp are both ubiquitously expressed? Again, one can evoke post-transcriptional regulation here, but given the striking mRNA expression overlap between chinmo and syp, such regulation (and thus large disconnect between mRNA and protein levels) would be rather striking. Resolving these questions seems fundamental to this study since, as it is presented now, the model is not supported by the presented data.

Please look above at our detailed response to point 1.

• The authors use Monocle to analyze the trajectory of tNBs, and find that "the states with highly expression of Imp were put at the beginning of the pseudotime". However, while Monocle is designed to find the transition trajectory/pseudotime based on transcriptome ordering, it cannot be used to determine the root or the base state of the trajectory. In addition, it is becoming common practice to seek independent analytical validation for such pseudotime analyses and in this regard, the "RNA velocity" tool (https://github.com/velocyto-team/velocyto.R) is a very exciting and intuitive approach. Finally, it would be valuable if staining of Chimo/Imp/Syp/E93 in one microscopy field would be performed to show that Chimo/Imp and Syp/E93 truly represent two different NB stages.

Concerning the identification of the root state, please look above at our detailed response to point 4.

We would love to be able to do a Chimo/Imp/Syp/E93 in one microscopy field. Unfortunately, this is currently impossible since two of the antibodies have been made in rabbits. However, our current pictures in Figure 1E and F convincingly show that all Chinmo and Imp overlap, that Chinmo/Imp-negative tNBs express Syp and that Chinmo and E93 expression pattern are complementary and not overlapping. Together, this shows that Chimo/Imp and Syp/E93 represents two different NB populations.

• The authors focus on cluster 7 and functionally validate its biological relevance. While functionally characterizing all clusters are beyond the scope of this manuscript, the author should at least discuss their significance. The cluster at the right side (Figure 1C, maybe cluster 1 or 8) for example is much separated from the other clusters. Which cells belong to this cluster? Any enrichment for specific biological processes? What can we learn here?

We now provide a more complete description of the clusters identified in the UMAP representation. In particular, we mention: “The presence of 4 large clusters (clusters 0, 1, 3, 4) within the E93 expression domain on the UMAP representation implies further cellular heterogeneity within the E93+ tNB population (Figure 1C, D). The two smaller clusters appear to be composed of tNBs expressing stress or growth arrest factors such as Gadd45, Irbp18, Xrp1 and GstE6 (Figure 1—figure supplement 2).”

Using Monocle, we clearly demonstrate that there are two main types of E93+ tNBs with different proliferative and metabolic properties (Figure 7 and 8 and Figure 8—figure supplement 2).

Reviewer #2:[…] Looking at the data from a developmental biology point of view, the findings may rather represent faulty neurogenesis in which decommissioning of neural stem cells and interfering with the temporal transcription factor cascades has not occurred properly, but that it appears that the cells in question can responds to the normal constraints of fly development in as much as it can when a key regulatory factor is removed. Removing more factors alters/worsens of course the situation.However, there are certainly valuable insights into fly specific developmental biology processes of neurogenesis and decommissioning of neuroblast divisions.

We thank reviewer 3 for the insightful thoughts on the manuscripts. Overall, we think that his impressions are right. Indeed, the predictability we observe, and that we have now strengthen in our revised version (Figure 2E), suggests that this type of tumors, with an early developmental origin, behaves more like developing tissues that somehow can’t fully terminate their developmental program, than like adult tumors that aberrantly activate signaling pathways because of the progressive accumulation of mutations and global genome instability. More generally, we think this might be a fundamental difference between pediatric tumors (usually carrying very few mutations) and adult tumors (usually carrying many mutations) (Grobner et al., 2018). For example, retinoblastoma is solely caused by the biallelic inactivation of RB1, while AT/RTs are caused by the biallelic inactivation of SMARCB1. Yet, despite the fact that retinoblastoma and AT/RTs are usually genetically stable, there is no doubt that they represent proper and highly aggressive tumors. We think this is also the case for the prosKD NB tumors we are investigating as it has been shown that they can be propagated for at least several months upon multiple transplantations in the abdomen of adult flies (Caussinus and Gonzalez, 2005, doi:10.1038/ng1632). We propose that transformation in such tumors with an early developmental origin is not due to genomic instability and accumulation of mutations but is rather due to the cooption of early developmental programs that favor embryonic-like proliferation.

More specifically, we identify here the coopted developmental program as being part of the larval NB-specific temporal patterning system. We show that the temporal patterning system is hijacked upon initial NB amplification due to the early loss of pros. Consequently, dysregulated temporal patterning creates a tissue that fails to ever stop growing and ultimately kills the fly (hence being considered as a tumor). We find that the temporal patterning is coopted with 100% penetrance if prosKD is induced during the correct early developmental window, generating reproducible tumors with predictable heterogeneity. Moreover, we find that two RNA-binding proteins Imp and Syp regulate the proliferative properties of the different types of tumor cells by regulating metabolic genes. Thus, coopted temporal patterning governs the growth, hierarchy and heterogeneity of NB tumors.

To our knowledge, this work provides the most detailed description of a hierarchical CNS tumor ever performed. Moreover, this work suggests that the presence of temporal patterning in neural stem cells makes the developing CNS particularly susceptible for cancer transformation. Given the recent demonstration that temporal patterning also exists in mammals (Telley et al. 2019), this could explain why CNS tumors are so prominent in children.